# Stochastic and Deterministic Processes Regulate Phytoplankton Assemblages in a Temperate Coastal Ecosystem

Dimitra-Ioli Skouroliakou,[a] Elsa Breton,[a] Solène Irion,[a] Luis Felipe Artigas,[a] Urania Christaki[a]

aUniversity Littoral Côte d'Opale, CNRS, Wimereux, France

**ABSTRACT** Assessing the relative contributions of the interacting deterministic and stochastic ecological processes for phytoplankton community assembly is crucial in understanding and predicting community organization and succession at different temporal and spatial scales. In this study, we hypothesized that deterministic and stochastic ecological processes regulating phytoplankton, present seasonal and repeating patterns. This hypothesis was explored during a 5-year survey (287 samples) conducted at a small spatial scale (~15km) in a temperate coastal ecosystem (eastern English Channel). Microscopy and flow cytometry quantified phytoplankton abundance and biomass, while metabarcoding data allowed an extended evaluation of diversity and the exploration of the ecological processes regulating phytoplankton using null model analysis. Alpha diversity of phytoplankton was governed by the effect of environmental conditions (environmental filtering). Temporal community turnover (beta diversity) evidenced a consistent interannual pattern that determined the phytoplankton seasonal structure. In winter and early spring (from January to March), determinism (homogeneous selection) was the major process in the phytoplankton community assembly. The overall mean in the year was 38%. Stochastic processes (ecological drift) prevailed during the rest of the year from April to December, where the overall mean for the year was 55%. The maximum values were recorded in late spring and summer, which often presented recurrent and transient monospecific phytoplankton peaks. Overall, the prevalence of stochastic processes rendered less predictable seasonal dynamics of phytoplankton communities to future environmental change.

**IMPORTANCE** While ecological deterministic processes are conducive to modeling, stochastic ones are far less predictable. Understanding the overall assembly processes of phytoplankton is critical in tracking and predicting future changes. The novelty of this study was that it addressed a long-posed question, on a pluriannual scale. Was seasonal phytoplankton succession influenced by deterministic processes (e.g., abiotic environment) or by stochastic ones (e.g., dispersal, or ecological drift)? Our results provided strong support for a seasonal and repeating pattern with stochastic processes (drift) prevailing during most of the year and periods with monospecific phytoplankton peaks.

**KEYWORDS** community assembly, phytoplankton, seasonality, ecological processes, coastal ecosystem, community ecology

Understanding the mechanisms that shape species' community structure is a central topic in ecology (e.g., references (1–3)). 'Niche theory' hypothesizes that species coexist due to their intraspecific and interspecific interactions and changing environmental conditions (4). On the contrary, 'neutral theory' assumes that all species are ecologically functionally equivalent, and species coexist due to random changes in the community structure because of stochastic processes of birth, death, colonization, extinction, and speciation (5). Given these two theories, Chesson (2) recognized that both niche and neutral processes act concomitantly in structuring communities. In line

Address correspondence to Dimitra-Ioli Skouroliakou, dimitra-ioli.skouroliakou@univ-littoral.fr.

The authors declare no conflict of interest.

with these perspectives, Vellend's conceptual framework (6) grouped four major ecological processes that drive species composition and diversity: selection, dispersal, speciation, and ecological drift. Selection refers to deterministic fitness differences between individuals of the same or different species, including environmental filtering and interactions among species (competition, predation, and facilitation). Dispersal is the movement of species across space, speciation is the generation of new genetic variation, and ecological drift represents random changes in species' relative abundance over time due to the inherent stochastic processes.

Assessing the relative importance of these processes has recently attracted the attention of microbial ecologists mostly in soil and freshwater environments by using concepts developed in terrestrial ecology (e.g., references (7–11)). These studies revealed the concomitant action of deterministic and stochastic processes in shaping communities. Deterministic processes are related to environmental conditions such as nutrient availability (9), species traits or interactions (reference (12) and references therein), while stochastic processes that include inherent randomness are less predictable and related to dispersal mechanisms, drift, and speciation (6). Most of the literature developing ecological theories (such as Hubbel et al. (3)) and the respective methodology derive from terrestrial ecology.

Historically, phytoplankton assemblages have been studied from a deterministic perspective based on their traits (e.g., (13, 14)), and their environment (15). However, deterministic processes in structuring phytoplankton communities (16) are insufficient to explain overall community structure and diversity patterns (17). Marine phytoplankton studies have been partly focused on stochastic (e.g., 18) or dispersal processes (e.g., (19)). Yet, there is a need to understand how deterministic and stochastic processes potentially change in one ecosystem at different time scales (11, 20). Two existing phytoplankton studies have quantified the relative contribution of both deterministic and stochastic processes focused on the large spatial scale (21) or short periods (less than a year) (22). However, the present study was the first to quantify both deterministic and stochastic phytoplankton assembly processes at a seasonal scale over a pluriannual sampling period.

Given the annual emergence of *Phaeocystis globosa* blooms and the high seasonal turnover in environmental conditions and phytoplankton, the eastern English Channel is a very suitable natural ecosystem to explore these ecological processes. It is a constantly mixed mesoeutrophic epicontinental sea, which has been undergoing anthropogenic disturbances since the last century (23). The nitrate enrichment with parallel silicate depletion in late winter promotes annual spring blooms of the Haptophyte *Phaeocystis globosa* (24, 25). Yet, after the wane of the *P. globosa* bloom, the seasonal succession is also marked by the increase in abundance and biomass of other planktonic taxa, such as heterotrophic bacteria, reaching high abundances (26), peaks of parasitic Syndiniales (27), diatoms (28), and dinoflagellates (29, 30).

Previous studies investigating drivers structuring planktonic communities in the eastern English Channel focused only on deterministic processes, such as intertaxon relations (31), environmental filtering (32), and predation (33). The present study aimed to explore the stochastic and deterministic ecological processes driving the seasonal organization of phytoplankton assemblages, and how these processes varied across seasons. We hypothesized that deterministic and stochastic ecological processes regulating phytoplankton, present seasonal and repeating patterns. For this, the seasonal diversity patterns combining morphological (i.e., counts by microscopy and flow cytometry) and metabarcoding (18S rRNA gene amplicon sequencing) data of phytoplankton obtained at five neighboring coastal stations in the eastern English Channel from 2016 to 2020 were investigated. Second, the phylogenetic structure (alpha diversity) and the phylogenetic turnover (beta diversity) in metabarcoding data using null models according to Stegen et al. (7, 34) were explored. Finally, considering the theoretical framework of Vellend (6), the relative importance of stochastic and deterministic ecological processes across seasons was quantified.

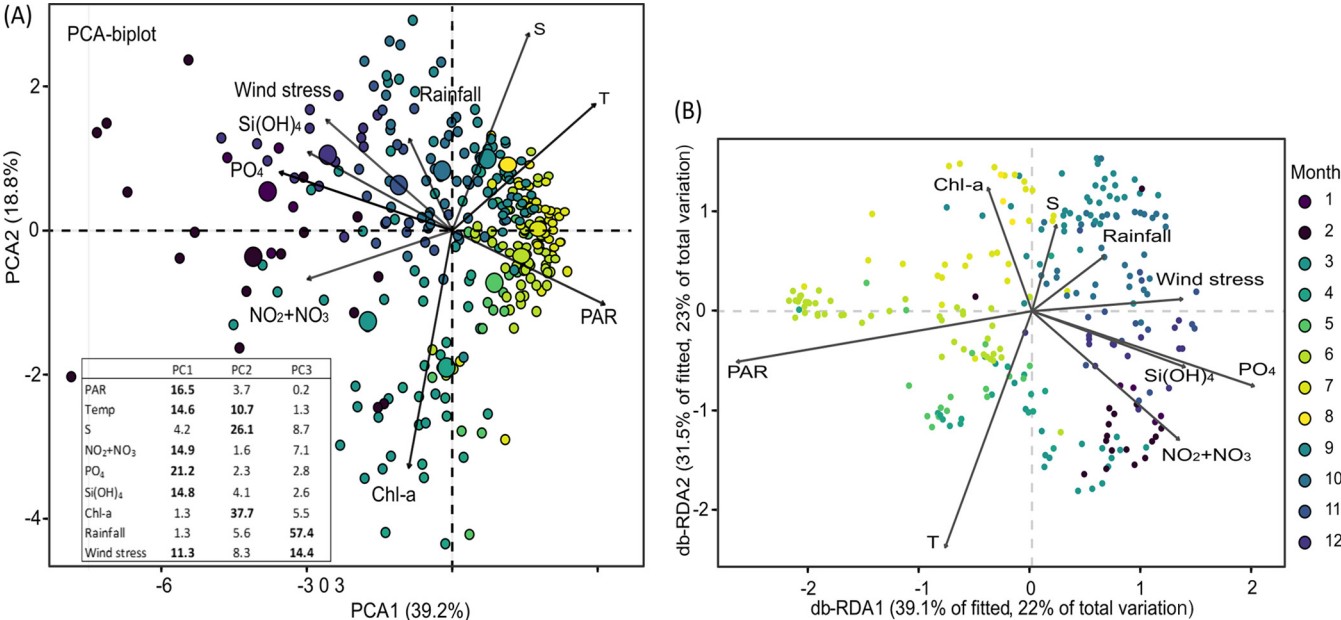

**FIG 1** (A) Principal-component analysis (PCA) illustrating the variations of the environmental variables (arrows) at all sampling dates (colored points with the size corresponding to the cos2 values of the PCA); photosynthetic active radiation (PAR, E m$^{-2}$ d$^{-1}$), temperature (T, °C), salinity (S, PSU), nitrite and nitrate (NO$_2$+NO$_3$ $\mu$M), silicate (Si[OH]$_4$, $\mu$M), phosphate (PO$_4$, $\mu$M), chlorophyll-a (Chl-a, $\mu$g L$^{-1}$), rainfall (Kg m$^{-2}$), wind stress (Pa). The table on the bottom left represents the percentage of the contribution of the different environmental variables in the building of the PCA axis. The most important contributors are in bold. (B) Distance-based redundancy (db-RDA) ordination illustrating the variations of the phytoplankton communities, based on metabarcoding data, (samples, colored dots) in relation to the environmental variables (black arrows) in the eastern English Channel at the DYPHYRAD and SOMLIT stations from March 2016 to October 2020.

## RESULTS

**Seasonality of the environmental variables and phytoplankton communities.**
Before the quantification of ecological processes regulating the phytoplankton communities across seasons, the seasonal variations of the environment and the phytoplankton communities were investigated. The environmental variables and phytoplankton community composition measured in the eastern English Channel at the Service d'Observation en Milieu Littoral (SOMLIT) and DYnamique PHYtoplanctonique le long de la RADiale (DYPHYRAD) stations evidenced clear seasonal patterns from December 2016 to October 2020 that were typical of temperate marine waters. Wind stress, nutrients, and chlorophyll showed great variability across seasons (Table S4 in Supplemental File 1). Nutrient inputs originated mainly from local rivers and reached relatively high values during fall and winter (Fig. S2A in Supplemental File 1). For example, silicate concentrations recorded in winter were high (e.g., on average 5.9 ± 1.9 $\mu$M in January), whereas values in spring and summer were low on average (e.g., 0.9 ± 0.9 $\mu$M in June; Table S4 and Fig. S2A in Supplemental File 1). The N/P molar ratio varied greatly across seasons (from 0.4 to 316), and most of the time strongly deviated from the Redfield ratio (35), (N/P = 16; Table S4 in Supplemental File 1). A comparison of the mean ranks (Kruskal Wallis and Nemenyi *post hoc* test) of environmental variables between the different stations revealed significant differences in salinity, phosphate, silicate, and Chl-a between the stations (Fig. S2B in Supplemental File 1). However, the environmental variables were of the same range and showed the same seasonal variation at all stations (Fig. S2B in Supplemental File 1). Principal component analysis (PCA) performed on the environmental data set showed that the first two principal components contributed to 58% of the total variance (Fig. 1A). The first principal component (PC1, 39.2%) was mainly formed, in decreasing order, by phosphate, PAR, nitrite and nitrate, silicate, temperature, and wind stress, opposing winter and summer conditions (Fig. 1A; Fig. S2A in Supplemental File 1). The second principal component (PC2, 18.8%) was mainly formed by decreasing order by Chl-a, salinity, and temperature associated with spring and autumn. Overall, summer and autumn samples formed

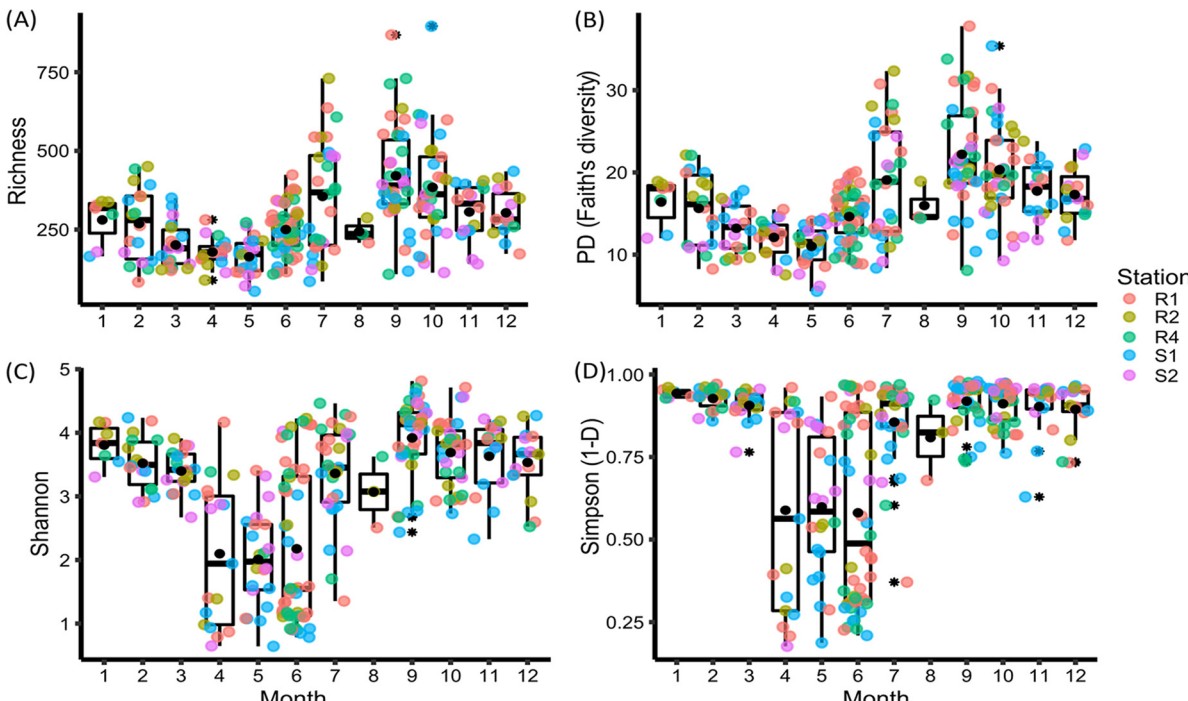

**FIG 2** Alpha diversity of the phytoplankton communities based on metabarcoding data collected in the eastern English Channel at the DYPHYRAD and SOMLIT stations from March 2016 to October 2020. (A) Richness, (B) PD, Faith's phylogenetic diversity, and the (C) Shannon and (D) Simpson (1-D) indices. Solid black lines represent the median, black dots the mean, colored dots the samples according to stations, and the black stars the outliers.

tighter groups on the PCA biplot than spring and winter samples, which were more dispersed (Fig. 2A).

The distance-based redundancy analysis (db-RDA) was applied to metabarcoding data of phytoplankton communities. The environmental variables showed that the first two axes explained 45% of the total variation in phytoplankton composition data (23% and 22% of the total variability for db-RDA1 and db-RDA2, respectively; Fig. 1B) with db-RDA1 and db-RDA2, highlighting a seasonal succession similar the PCA. The permutation test showed that PAR and temperature mainly contributed to the overall variability by 21% ($P = 0.001$) and 18% ($P = 0.001$), respectively, followed by nutrients (nearly 10%, $P = 0.01$; Table S5 in Supplemental File 1). Our study was one of the rare ones considering both morphological and metabarcoding data using a relatively large data set (287 samples). Accordingly, it was important to confront the two data sets and see if they showed similar trends because metabarcoding data were subjected to PCR biases and were always expressed in relative abundances while morphological data were absolute abundances. For this, the db-RDA was also applied to the microscopy data, which revealed similar seasonal trends (Fig. S3 in Supplemental File 1). The first two axes showed that the selected environmental variables explained nearly 37% of the total variation (db-RDA1, 24%; db-RDA2, 13%) in phytoplankton data. The permutation test showed that PAR (24%, $P = 0.001$), temperature (10%, $P = 0.001$), and nutrients (10%, $P < 0.05$) mainly contributed to the overall variability (Table S6 in Supplemental File 1).

**Phytoplankton community inferred with metabarcoding and morphological data.** Phytoplankton communities showed high variability in alpha diversity (Fig. 2A to D). The overall trend observed was decreasing richness, phylogenetic Faith's diversity (PD), and Shannon index from winter to spring, with the lowest observed values during the *P. globosa* spring bloom (April, May) and highest observed values in summer and autumn. Accordingly, the Simpson diversity index (1-D) showed the lowest values during the *P. globosa* spring bloom and relatively high values during the rest of the year (Fig. 2A to D).

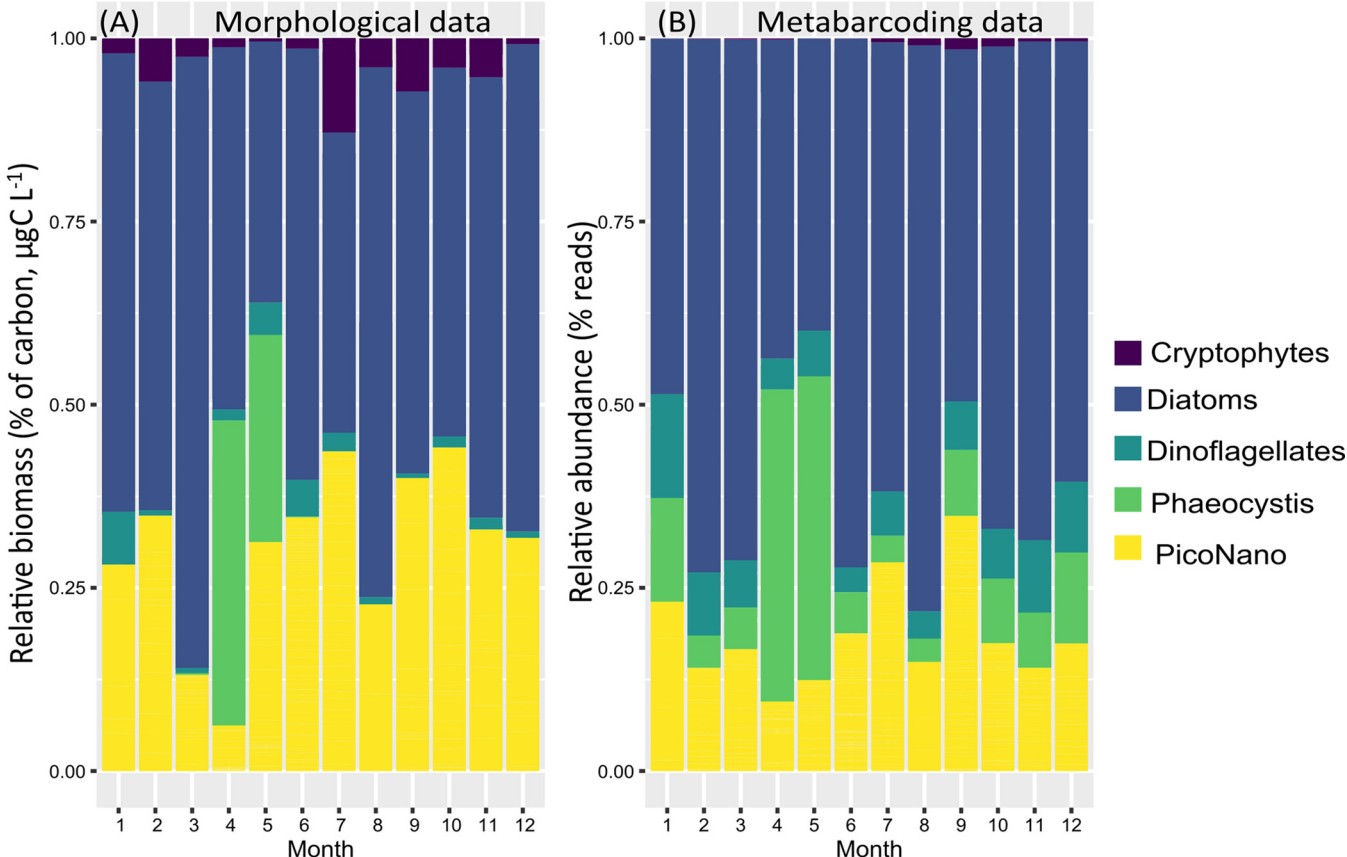

**FIG 3** Phytoplankton community structure in the eastern English Channel at the DYPHYRAD and SOMLIT stations from March 2016 to October 2020. (A) From microscopy and cytometry (relative biomass as a percentage of carbon, see Table S3 in Supplemental File 1 for biomass calculation). (B) From rarefied metabarcoding data (relative abundance as the percentage of reads of the ASVs).

Based on morphological data (i.e., microscopy and cytometry counts), diatoms dominated the phytoplankton biomass (mean 55%) across all seasons except April to May, when the haptophyte *P. globosa* increased in biomass (42% and 28%, respectively, Fig. 3A). pico-nano-phytoplankton (PicoNano) contributed 32% of the total biomass, while cryptophytes and dinoflagellates accounted for only 4% and 3% of the total phytoplankton biomass, respectively (Fig. 3A). Metabarcoding data also reflected the dominance of diatoms as relative read abundance (mean 60%) in the community and the *Phaeocystis* bloom in April and May (43% and 41%, respectively). PicoNano, cryptophytes, and dinoflagellates contributed 20%, 0.5%, and 6% of the mean relative read abundance, respectively (Fig. 3B). Furthermore, no significant differences were found among the stations for the number of reads, cell counts, and biomass of the different phytoplankton groups recorded in this study (Fig. S4 in Supplemental File 1).

Besides the *P. globosa* bloom in April and May (Fig. 4A), several important peaks belonging to different groups were observed in spring, summer, and autumn, in both data sets. In July 2016, the planktonic diatom *Chaetoceros socialis* reached a maximum value of $3.1 \times 10^6$ cells L$^{-1}$ (Fig. 4B, Fig. S5 in Supplemental File 1), which coincided with a relatively high number of *Chaetoceros* reads (32%, Fig. S5 in Supplemental File 1). In July 2017, *Guinardia* showed a relatively high number of reads (69%, Fig. S6 in Supplemental File 1) in contrast to low abundance ($18 \times 10^3$ cells L$^{-1}$) in microscopy data. In June 2018, a peak of the pennate diatom *Pseudonitzschia pungens*, reached $4.8 \times 10^6$ cells L$^{-1}$ (Fig. 4B). The transient *P. pungens* peak was also clearly observed in metabarcoding data, reaching 73% of relative read abundance (Fig. S7 in Supplemental File 1). In addition, the centric diatom *Leptocylindrus danicus* marked diatom community structure also in June with a maximum concentration of $1.5 \times 10^6$ cells L$^{-1}$ in 2018 (Fig. 4B;

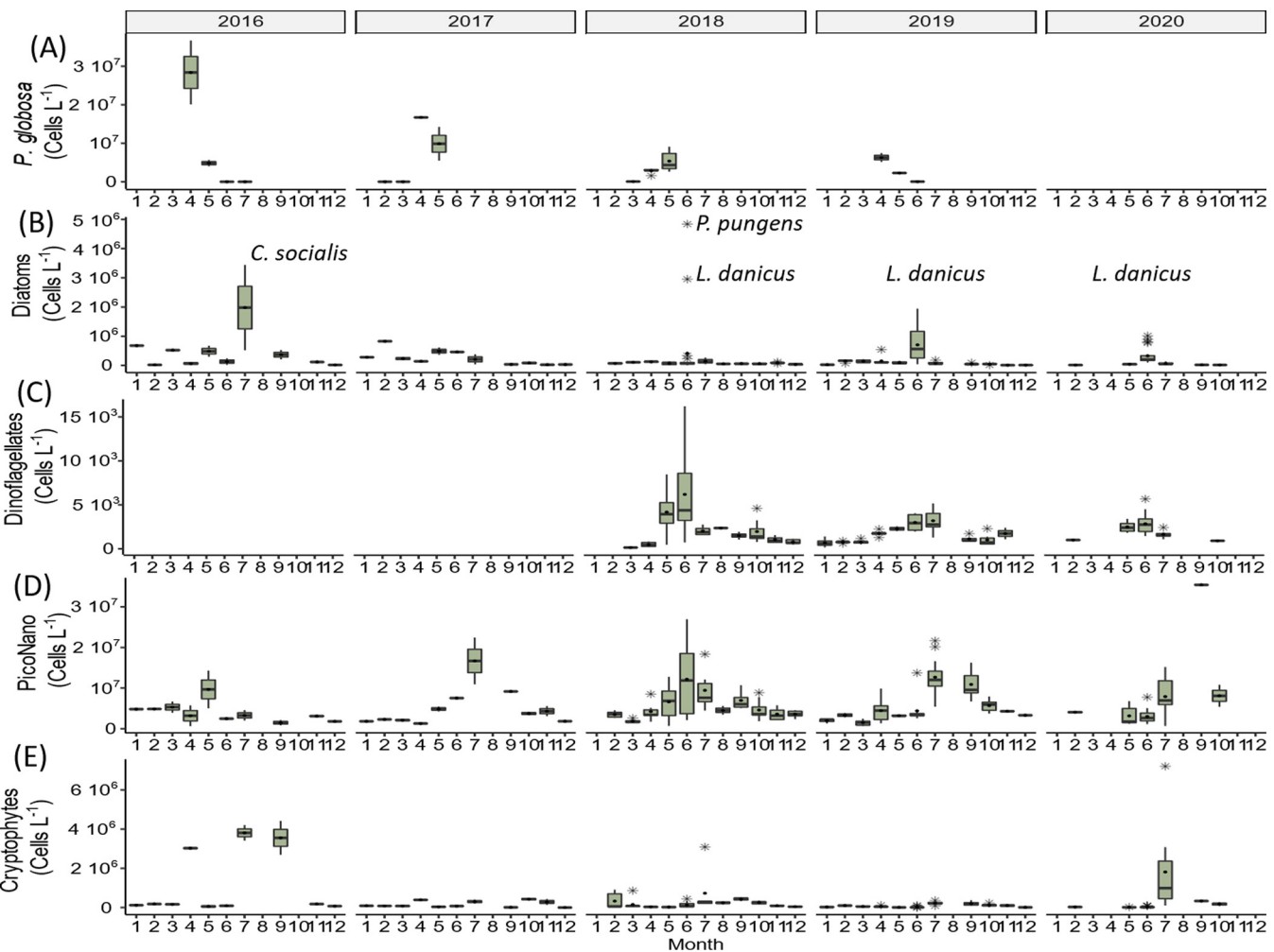

**FIG 4** Abundance (cells L$^{-1}$) of the phytoplankton groups identified in the eastern English Channel at the SOMLIT and DYPHYRAD stations from March 2016 to October 2020 based on microscopy and flow cytometry data. (A) *P. globosa*, (B) diatoms, (C) dinoflagellates (Gymnodinium and Prorocentrum), (D) PicoNano, and (D) cryptophytes. No data were available from 2016 to 2017 for dinoflagellates and all phytoplankton from February 14, 2020 to May 20, 2020 because of the sanitary crisis). Solid black lines represent the median, black dots the mean, and the black stars the outliers.

Fig. S7 in Supplemental File 1) and was also abundant in June of 2019 and 2020 (Fig. 4A, Fig. S8 and S9 in Supplemental File 1). Dinoflagellates showed a peak in June 2018 (13.5 × 10$^3$ cells L$^{-1}$, Fig. 4C) attributed to *Prorocentrum minimum*, which contributed up to 9% of the total relative abundance of reads (Fig. S7 in Supplemental File 1). PicoNano showed peaks of abundance generally in spring and summer, while the maximum abundance was recorded in September 2020 (35.4 × 10$^6$ cells L$^{-1}$, Fig. 4D). According to metabarcoding, PicoNano was dominated by the coccolithophorid *Emiliania* and the nanoplanktonic diatom *Minidiscus*. Several peaks were observed in cryptophyte abundance in July 2016, 2018, and 2020. In 2016, peaks were also observed in April and September (Fig. 4E). Based on metabarcoding, the dominant cryptophyte was assigned to *Plagioselmis*.

**Phylogenetic structure, temporal turnover, and ecological processes driving the phylogenetic phytoplankton community structure.** Mantel correlograms correlating the phylogenetic distances to the niche distances at different distance classes detected significant positive correlations across short phylogenetic distances (<0.4 phylogenetic distance; Fig. S10 in Supplemental File 1). Except for salinity, all the evaluated environmental variables showed significant negative correlations over intermediate phylogenetic distances. These results supported that phylogenetic metrics can be applied to infer ecological assembly processes. Significant positive correlations across long phylogenetic distances were observed for PAR (nearly 0.8 phylogenetic distance; Fig. S10 in Supplemental File 1).

**TABLE 1** Definitions of the different assembly processes, and respective model conditions referenced from Webb et al. (44), Stegen et al. (7, 34), and Zhou and Ning (12)

| Process | Deterministic processes | | Stochastic processes | |
|---|---|---|---|---|
| **Diversity** | **Alpha diversity** | | | |
| Clustering | Phylogenetic clustering | Phylogenetic overdispersion | | |
| Definition | Environmental conditions, selecting those species capable of survival and persistence in a local environment (environmental filtering) | Competitive exclusion, resulting in limiting similarity (overdispersion) | | |
| Phylogenetic structure index | $NRI > 0$ | $NRI < 0$ | | |
| **Diversity** | **Beta diversity** | | | |
| Selection, Dispersal, Drift | Homogeneous selection | Heterogeneous selection | Dispersal limitation | Homogeneous dispersal | Drift |
| Definition | Consistent environmental factors cause a low compositional turnover | Shifts in environmental factors cause a high compositional turnover | Movement of an individual was restricted | High rate of movement of an individual from one location to another | Population size fluctuates due to chance events |
| Phylogenetic turnover index | $\beta NRI < -2$ | $\beta NRI > 2$ | $-2 < \beta NRI < 2$ | | |
| Taxonomic turnover index | - | - | $RC_{bray} > 0.95$ | $RC_{bray} < -0.95$ | $-0.95 < RC_{bray} < 0.95$ |

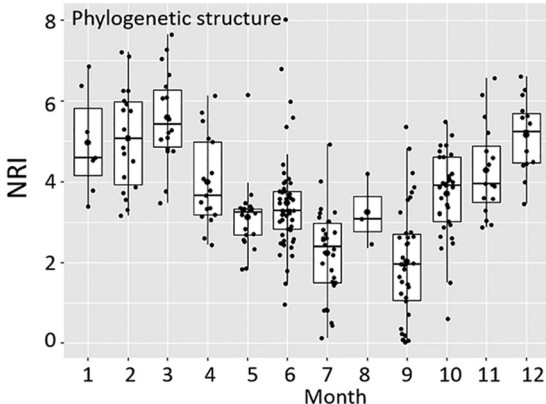

**FIG 5** Phylogenetic structure (alpha diversity) of the phytoplankton community in the eastern English Channel at the DYPHYRAD and SOMLIT stations from March 2016 to October 2020 based on metabarcoding data. Phylogenetic structure based on the net relatedness index (NRI) with NRI > 0 and NRI < 0 suggested phylogenetic clustering and overdispersion, respectively (see Table 1). Solid black lines represent the median and black dots the mean.

Ecological processes governed the alpha diversity of phytoplankton (i.e., phylogenetic structure) were investigated using the net relatedness index (NRI). It showed that phylogenetic clustering prevailed during all seasons suggesting environmental filtering (NRI > 0, Table 1). The strongest values of phylogenetic clustering were detected from January to March (NRI > 3, Fig. 5). From April to July, NRI values showed a decreasing trend, indicating a tendency toward a weaker phylogenetic clustering at this time of the year. From September to December, NRI values showed again an increasing trend (Fig. 5).

To test if a phylogenetic structure (i.e., NRI) and turnover (i.e., $\beta$NRI) were attributed to different environmental variables a permutational multivariate analysis of variance (PERMANOVA) was applied. PAR and temperature were the only variables significantly linked with phytoplankton phylogenetic structure and phylogenetic turnover (Table S8 and S9 in Supplemental File 1; $P < 0.001$), although they explained a very small amount of the variance of the data set. Most of the variance remained unexplained with residuals presenting 67% and 87% for the phylogenetic structure and phylogenetic turnover (Table S8 and S9 in Supplemental File 1).

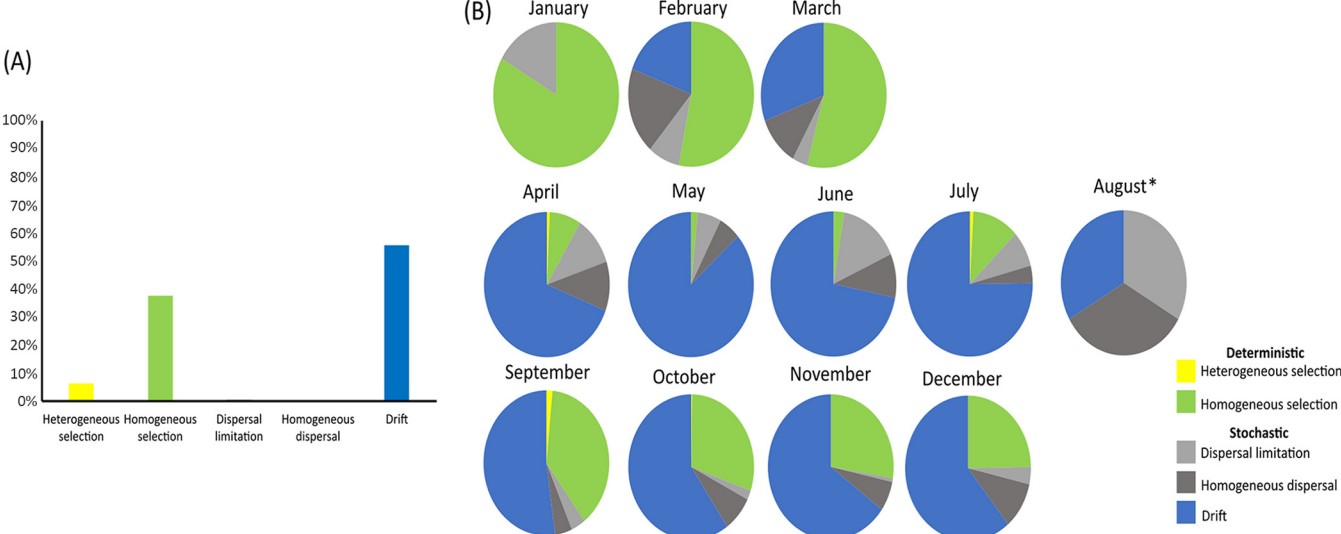

**FIG 6** The relative importance of the ecological processes (beta diversity) driving phytoplankton communities in the eastern English Channel at the DYPHYRAD and SOMLIT stations from March 2016 to October 2020. A: considering the whole data set, B: data discriminated per month. *, only three samples were available for August, so the data was not interpretable.

**TABLE 2** Summary of the seasonal characteristics of the phytoplankton community in coastal waters of the eastern English Channel[a]

| | Median in entire dataset | January-March | April-July | September-December |
|---|---|---|---|---|
| Dominant phytoplankton (morphological and metabarcoding) | | *Thalassiosira* sp. | *Phaeocystis globosa, Chaetoceros socialis, Leptocilyndrus danicus, Pseudonitzschia pungens, Prorocentrum minimum,* PicoNano | *Thalasiossira* sp., *Ditylum sp.,* piconanophytoplankton |
| PAR (E m$^{-2}$ d$^{-1}$) | 60.8 | Low (35.5) | High (84.9) | Low (41.7) |
| T (°C) | 15 | Low (7.7) | Moderate (15.2) | Moderate (16.5) |
| NO$_2$+NO$_3$ | 1.0 | High (4.3) | Low (0.5) | Moderate (1.3) |
| Si(OH)$_4$ | 1.3 | High (2.9) | Low (0.8) | Moderate (1.8) |
| PO$_4$ | 0.2 | High (0.4) | Low (0.1) | Moderate (0.2) |
| Wind stress (Pa) | 0.04 | High (0.1) | Moderate (0.04) | High (0.1) |
| Rainfall (Kg m$^2$) | 1.3 | Moderate (1.1) | Low (0.8) | High (2.0) |
| Richness | 272 | Low (213) | Low (218) | High (361) |
| Taxonomic alpha diversity (Shannon) | 3.4 | High (3.5) | Low (2.5) | High (3.8) |
| Ecological processes (Alpha diversity) | 1.4 | Environmental filtering (3.1) | Environmental filtering (0.8) | Environmental filtering (1.5) |
| Ecological processes (Beta diversity) | | Homogeneous selection | Drift | Drift+Homogeneous selection |

[a]The median values are given in parentheses. The relative terms "High," "Moderate," and "Low" refer to the median of the entire data set.

The different ecological processes, selection, dispersal, and drift were quantified using the null model analysis. The analysis applied to the entire data set showed that drift and homogeneous selection were the dominant ecological processes driving the seasonal succession of the phytoplankton community, contributing to 55% and 38%, respectively, over the entire period of study (Fig. 6A). Heterogeneous selection contributed weakly to the seasonal succession of the phytoplankton community (6%), and dispersal did not have any significant influence. The null model analysis applied additionally at the monthly scales showed three distinct patterns. First, phytoplankton succession in winter-early spring was dominated by homogeneous selection, contributing to 83%, 52%, and 54% of the assembly processes in January, February, and March, respectively. Second, spring and summer periods were mostly dominated by drift mechanisms (contributing from 69% to 87%, depending on the month). Third, autumn was dominated primarily by drift, contributing from 52% to 65% to the assembly processes with homogeneous selection accounting from 25% to 38% (Fig. 6B; Table S7 in Supplemental File 1).

## DISCUSSION

Here, microscopy and flow cytometry allowed phytoplankton biomass to be quantified, while metabarcoding data provided an extended evaluation of its diversity. Alpha diversity of phytoplankton communities was regulated by environmental filtering. Drift, followed by homogeneous selection, were the major mechanisms regulating the temporal turnover in community composition (beta diversity) and prevailed across seasons. Three periods were evidenced, including (i) winter-early spring, with homogeneous selection as the major process regulating the phytoplankton communities, composed mainly of diatoms communities (e.g., *Thalassiosira*); (ii) spring-summer, with drift as the major process in community assembly during the bloom of *P. globosa* and during the transient peaks of various taxa (diatoms, dinoflagellates, and pico-nanophytoplankton); and (iii) autumn, with a combination of drift and homogeneous selection as a major ecological process in phytoplankton community assembly dominated by diatoms (Table 2). Overall, we evidenced that deterministic and stochastic ecological processes varied across seasons in alpha and beta diversity.

**Seasonal diversity patterns.** Overall, the relative abundance of the major phytoplankton groups inferred by metabarcoding was in good accordance with the relative carbon biomass inferred by morphological approaches for PicoNano and cryptophytes and to a lesser degree for diatoms (36) (Fig. 3). Moreover, the distance-based RDA analysis showed similar seasonal patterns in both data sets (Fig. 1B, Fig. S3 in Supplemental File 1).

The most abundant diatom taxa were common in both data sets. Three diatom taxa

showed transient blooms in summer (June-July). The chain-forming centric diatom *L. danicus* had been previously reported in high abundances in the sampling area during summer (28, 37). A good correspondence between a high number of reads and cell counts of *Leptocylindrus* has been reported also in the Gulf of Naples (38). The chain-forming diatom *Pseudonitzschia* is known to form dense blooms along the French coast of the eastern English Channel and is often a co-occurring species of the *P. globose* (reference (39) and references therein). The colony-forming diatom *C. socialis* observed in July has been reported in the English Channel in spring and summer (28, 40).

Dinoflagellates are known to be overrepresented in sequencing data, and their use in numerical analysis can lead to important biases (e.g., references (41, 42)). In this study, two small mixotrophic dinoflagellates (*Gymnodinium* and *Prorocentrum*) were included in the analysis because they were the only dinoflagellates to exhibit relatively high abundances in microscopy data, while they did not represent an exaggerated number of reads in meta-barcoding data (Fig. 3, S4). *Gymnodinium* and *Prorocentrum* were prominent members of the protist community in previous studies in the eastern English Channel (33, 43). The genus *Gymnodinium* showed relatively stable cell numbers across seasons, while the species *P. minimum* showed a peak in 2018. Intense but brief blooms of *P. minimum* have been previously reported during the summer months in the Western English Channel (43). The *P. globosa* bloom in April and May was clear in both data sets. However, metabarcoding data evidenced the presence of low relative abundances of *P. globosa* all year long (Fig. 3B), which were not recorded by microscopy and cytometry because of its very low abundance. Piconanophytoplankton showed maximum concentrations also in summer reaching $2 \times 10^7$ cells L$^{-1}$, and one in fall $3.5 \times 10^7$ cells L$^{-1}$ (43) (Fig. 4D).

**Ecological processes shaping phytoplankton seasonal organization.** The NRI index calculated for the whole phytoplankton community evidenced that the alpha diversity of phytoplankton communities was governed by environmental filtering (44) (Table 1 and 4; Fig. 5). Environmental filtering was the effect of environmental conditions selecting those species capable of survival and persistence in a local environment (45, 46), and it is known to play a major role in structuring marine phytoplankton communities (47–50). Phytoplankton communities, based on the phylogenetic temporal turnover (beta diversity), were assembled across seasons through a concomitant action of deterministic and stochastic processes. However, stochastic processes (i.e., drift) contributed far more than ecological selection to community assembly, except for winter and early spring (from January to March), when homogeneous selection regulated phytoplankton communities (Fig. 5 and 6).

The dominance of homogeneous selection in winter and early spring was coherent with the strong environmental filtering conditions, such as the high nutrient concentration values, and low light availability recorded during this period (NRI > 4; Fig. 5A, see also Table 1 and 2). Hence, homogenous selection has been seen as the selection of species with common and appropriate genomic architecture and metabolic strategies for surviving and persisting in a local environment (e.g., 51, 52) implying an increase in community similarity (53). This process dominated community assembly when environmental conditions were spatially homogenous (Fig. S2B in Supplemental File 1) (20, 53). In the geographic scale of our study (ca 15 km), the coastal waters of the eastern English Channel represented a homogeneous pool of phytoplankton taxa undergoing similar selection processes (Fig. 6, S4). The dominance of diatoms at this time of the year, mainly of the genus *Thalassiosira*, was coherent with the worldwide observations of diatoms thriving in light-limited, nutrient-enriched, and colder waters submitted to relatively high wind-driven turbulence (54, 55). Silica frustule and large centric vacuole are considered key traits for diatom success in winter and early spring for protecting against mechanical and haline shocks (56, 57) and optimizing light affinity (58).

The prevalence of stochastic processes in phytoplankton community assembly in beta diversity was observed during the rest of the year (from April to December; Fig. 6 and Table 2). Maximum drift values were recorded in late spring and summer periods that presented monospecific phytoplankton peaks. This was coherent with the large quantity of unexplained variance between environmental variables and the phytoplankton community

in terms of abundance, phylogenetic structure, and turnover (db-RDA and PERMANOVA, respectively). The dominance of drift was in accordance with a previous study quantifying the ecological processes in natural ecosystems using the same analytical approaches as the present study (59). These authors found that microeukaryotic communities were governed by drift (72%), while the relative contribution of selection and dispersal was low.

However, identifying the underlying mechanisms and factors favoring stochasticity was challenging. Hence, studies are still scarce, and detecting stochastic processes may suggest the lack of consideration of unmeasured environmental variables (12), as well as from a mixture of antagonistic processes (60). Moreover, multiple factors may influence the relative importance of stochastic versus deterministic processes in community assembly, including predation (61), productivity (62), community size (63), resource availability (64), as well as a disturbance (65). For example, predators can increase the importance of stochastic processes by reducing the number of individuals that can live in each environment and, thus, the community size by increasing the probability of species going extinct locally (61). Nonetheless, the minor role played by dispersal in shaping phytoplankton communities in this study area (Fig. 6) was coherent with the small spatial scale investigated (~15 km) (e.g., references (66, 67)).

In this study, the continuous decrease in species richness observed from February to May (Fig. 2A) potentially suggests intense stress for the diatom community which, despite increasing light, faces silicate limitations during this period (Fig. S2A in Supplemental File 1) (25). This presumption of stress was reinforced by the low degree of silicification, and the decrease of the functional evenness observed by Breton et al. (32) in May. Note that functional evenness describes the evenness of abundance distribution in functional trait space (68). A decrease in functional evenness, which reflects underused parts of the niche (68), is considered a fingerprint of disturbance (69). *P. globosa*, although it is a poor competitor for nitrate and has a lower maximum growth rate than diatoms (28) does not require silicate and thus it blooms under limiting silicate and excess nitrate in spring (70). *Phaeocystis* has also a strong protection against grazing by forming colonies and may further benefit from the increased grazing of mesozooplankton on the microzooplankton, potential predators of *Phaeocystis* (71).

Despite the relatively low nutrient levels after the *P. globosa* bloom, phytoplankton richness and PD values increased (Table 2) and several transient peaks belonging to different phylogenetic groups (diatoms, dinoflagellates, and PicoNano) appeared in summer (Fig. 4) (27). This suggests that other mechanisms lowered the competitive exclusion in shaping phytoplankton communities at this time of the year. One plausible explanation is that the new niche opportunities that might have resulted from intense bacteria activity remineralizing the organic resource derived from *P. globosa*. This resource is recurrently released in May in the seawater (26), and/or from the underused and vacant niches left open after the species loss at the end of the *Phaeocystis* bloom. Such a large input of dissolved organic material may be compared to large inputs of nutrients, which is typically considered a perturbation that favors ecological drift (5) through the enhanced growth of a variety of species and by reducing competition. Jurburg et al. (72) stated that ecological drift in soil microbial communities was due to niche enlargement after a perturbation. Mixotrophy typically reflects such a possibility (21). The dinoflagellate *P. minimum* showed that the peak in abundance in June was mixotrophic (Fig. 4B) (73). This species grows photosynthetically on inorganic nutrients but compensates for low concentrations of inorganic nitrogen by mixotrophic utilization of organic nitrogen and other compounds released at the end of *P. globosa* bloom. Other alternative strategies also exist, such as adaptation to high light levels. Indeed, the diatoms *L. danicus* and *P. pungens* are considered adapted to high light conditions (43, 74), whereas PicoNano and cryptophytes, can acquire nutrients in low concentration; outcompeting larger cells for nutrient uptake (75) due to the high surface/volume ratio reducing the "package effect" (76), and increasing nutrient diffusion (77), compared to larger cells. Overall, these specific adaptations provide a better efficiency for growth that is necessary for maintenance and ecological success during seasons with low nutrient and high light levels.

Autumn phytoplankton communities were subjected to the combined action of drift and selection processes. The local environment progressively increased the action of environmental filtering and, consequently, the contribution of homogeneous selection to community assembly increased (Fig. 5 and 6, and Table 2). Nutrients became progressively available again, light and temperature diminished, while high turbulence values (wind stress as a proxy) enhanced physical mixing (Fig. S2 in Supplemental File 1). This could explain the high richness, diversity, and community characterized by large diatoms such as *Thalassiosira* sp. and *Ditylium* sp. (e.g., Fig. S7 in Supplemental File 1). Drift remained a major mechanism in autumn, potentially related to high grazing pressure in late summer-autumn (33). External forces such as wind stress and salinity showed high variability and extreme values particularly in September (Fig. S2 in Supplemental File 1), which seemed to be a transitional period between summer and autumn conditions (Fig. 1A; Fig. S2 and S3 in Supplemental File 1).

We acknowledge that there are several limitations to discuss. Inferring ecological processes based on phylogenetic metrics was challenging. The phylogenetic signal was required, which was detected in our study (Fig. S10 in Supplemental File 1). The phylogenetic signal has been confirmed in natural phytoplankton communities and based on evolutionary models (78), but the opposite has been demonstrated in an experimental study focused on eight species of freshwater green algae (79). However, our results present the overall action of ecological processes at the whole community level, and not on a particular taxonomic group. Different taxonomic classes may be structured by different processes (21). The sampling effort was also important. For example, in this study, only three samples were available for August. Thus, the results could not be interpreted (Fig. 6). Finally, the biases derived from PCR and sequencing may add bias in calculating the importance of the ecological processes in community assembly. Nonetheless, in this study, there was, which was discussed above. a relatively good correspondence between morphological and metabarcoding data. Molecular and morphological data are complementary but unfortunately are rarely considered together in actual marine planktonic studies, and this was one of the strong points of our work.

Concluding this study, null modeling based on phytoplankton metabarcoding data revealed that stochastic and deterministic processes present seasonal and repeating patterns. Our results provided strong support that, except for winter and early spring, the ecological drift prevailed during the rest of the year and the periods presented by recurrent and transient monospecific phytoplankton peaks. The prevalence of stochastic processes renders *a priori* the seasonal dynamics of phytoplankton communities less predictable. In this context, the exploration of ecological processes driving phytoplankton communities in the long term is critical in our understanding of pelagic ecosystems' response relative to environmental variability.

## MATERIALS AND METHODS

**Sampling strategy.** Subsurface seawater (2 m depth) of five coastal stations along a ca. 15 km transect was sampled from March 2016 to October 2020 in the eastern English Channel (Fig. S1 in Supplemental File 1). During, the first 2 years, the S1 inshore station and the S2 offshore station of the SOMLIT monitoring network (https://www.somlit.fr/) were sampled on a bi-weekly basis. From 2018 to 2020 three stations were added: the coastal stations R1, R2, and the offshore station R4 from the local monitory transect DYPHYRAD. The sampling was carried out weekly and more intensively for some periods between 2018 and 2020. A total of 322 samples were gathered during 169 sampling campaigns at these five stations (Table S1 and S2 in Supplemental File 1).

**Environmental variables.** Sea surface temperature (T, °C) and salinity (S, PSU) were measured *in situ* with a CTD Seabird profiler. The average subsurface daily PAR experienced by phytoplankton in the water column for 6 days before sampling was obtained from global solar radiation (GSR, Wh m$^{-2}$) as described in Breton et al., (32). Wind stress (Pa) was calculated as described in Smith (80). In addition, seawater macronutrient concentrations were analyzed according to Aminot and Kérouel (81). Chlorophyll-a (Chl-a) concentration was measured by fluorometry as described in (82). Additional details on environmental data acquisition and sample analysis can be found at https://www.somlit.fr/en/ and in Supplemental File 1.

**Phytoplankton microscopic and cytometric counts (morphological data).** For diatoms and *P. globosa* counting, 110 mL water samples were collected and fixed with Lugol's-glutaraldehyde solution (1% vol/vol). For dinoflagellates, another 110 mL was fixed with acid Lugol's solution (1% vol/vol) (data for dinoflagellates were available from February 16, 2018; Table S2 in Supplemental File 1). Phytoplankton was examined, when possible, to the species or genus level using an inverted

microscope (Nikon Eclipse TE2000-S) at ×400 magnification after sedimentation in a 10, 50, or 100 mL Hydrobios chamber, as described previously in Breton et al. (32).

The abundance of picophytoplankton and nanophytoplankton (PicoNano,0.2 to 20 $\mu$m) and cryptophytes were enumerated by flow cytometry with a CytoFlex cytometer (Beckman Coulter). For all samples, 4.5 mL was fixed with paraformaldehyde (PFA) at a final concentration of 1% and stored at −80°C until analysis (83). Phytoplankton cells were detected according to the autofluorescence of their pigments (Chl-a, Phycoerythrin). *P. globosa* was discriminated from the PicoNano group based on orange fluorescence (Phycoerythrin, 496 nm).

Biovolumes for diatoms and cryptophytes, dinoflagellates, and *P. globosa* were estimated using an image analyzer system and standard geometric forms according to (84). Then, the carbon-biovolume relationships were estimated following prior studies (85–87) (see Table S3 in Supplemental File 1 for details).

**DNA barcoding.** For DNA extraction 4 to 7 L of seawater was filtered on 0.2 $\mu$m polyethersulfone (PES) membrane filters (142 mm, Millipore, U.S.A.) after a prefiltration step through 150 $\mu$m nylon mesh (Millipore, U.S.A.) to remove metazoans. All filters were stored at −80°C for 18S rRNA genes amplicon Illumina MiSeq sequencing. A quarter of the PES filter was used for DNA extraction following the DNAeasy PowerSoil Pro kit (Qiagen, Germany) manufacturer's protocol. The 18S rRNA gene V4 region was amplified using EK-565F (50-GCAGTTAAAAAGCTCGTAGT) and UNonMet (50-TTTAA GTTTCAGCCTTGCG) primers (88). Pooled purified amplicons were then paired-end sequenced on an Illumina MiSeq 2 × 300 platform. Quality filtering of reads, identification of amplicon sequencing variants (ASV), and taxonomic affiliation based on the PR2 database were done in the R-package DADA2 (89, 90).

A total number of 41,179 ASVs were identified from 6,366,087 reads in 287 samples, containing Metazoa, Streptophyta, Excavata, Alveolata, Amoebozoa, Apusozoa, Archaeoplastida, Hacrobia, Opisthokonta, Rhizaria, and Stramenopiles. For this study, only ASVs affiliated with Hacrobia, Stramenopiles, and Archaeoplastida were kept. In addition, mixotrophic dinoflagellates ASVs related to *Gymnodinium* and *Prorocentrum* were considered for their ecological importance. *Gymnodinium* is known to dominate dinoflagellate abundance and biomass in the eastern English Channel (33), while *Prorocentrum* showed a momentary increase in this study. Unaffiliated eukaryotic ASVs, singletons, and doubletons were removed, obtaining a final phyloseq object containing 6,471 ASVs corresponding to 2,448,955 reads, in 287 samples. The data set was rarefied to the lowest number of reads (1,020), resulting in 274,380 reads corresponding to 4,141 ASVs from 269 samples.

**Diversity, statistical, and community assembly analyses.** To describe the phytoplankton communities, the alpha diversity and Faith's phylogenetic diversity indices were calculated. Kruskal-Wallis and the *post hoc* Nemenyi test were used to test if phytoplankton groups significantly differ among stations. To explore and summarize seasonal variations in the abiotic environment the multivariate principal component analysis (PCA) was performed. Distance-based RDA analysis was applied considering morphological and metabarcoding data. All analysis was performed in R (91).

To infer ecological assembly processes governing alpha and beta diversity, null models based on metabarcoding data were applied according to the framework by Stegen et al. (7, 34) and reviewed by Zhou and Ning (12). All tests presented here were applied according to this framework. Briefly, the community assembly analysis was based on the comparison of observed community turnovers (shifts in composition across samples), phylogenetic turnovers (shifts in composition weighted by the phylogenetic similarity between taxa), and turnovers expected by chance (in null models), to estimate whether the differences between pairs of communities were explained by dispersal, selection, or ecological drift. According to phylogenetic community composition (alpha diversity) within each sample, the phylogenetic structure was divided into environmental filtering and overdispersion, which were deterministic processes (Table 1). Based on phylogenetic turnover in community composition (beta diversity), deterministic processes were divided into homogeneous selection (i.e., consistent environmental factors cause low compositional turnover) and heterogeneous selection (i.e., high compositional turnover caused by shifts in environmental factors). However, stochastic processes were divided into homogeneous dispersal (i.e., low compositional turnover caused by high dispersal rates), dispersal limitation (i.e., high compositional turnover caused by a low rate of dispersal), and ecological drift that can result from fluctuations in population sizes due to chance events (7, 34, and 47; Table 1). The implicit hypothesis was that phylogenetic conservatism exists, which means that ecological similarity between taxa was related to their phylogenetic similarity (i.e., phylogenetic signal) (39). Mantel correlograms were applied to detect phylogenetic signals, which correlate the phylogenetic distances to the niche distances at different distance classes (e.g., references (9, 92); further information in Supplemental File 1). Significant positive correlations indicate that ecological similarity among ASVs was higher than expected by chance within the distance class. Alternatively, significant negative correlations indicated that ASVs were more ecologically dissimilar than expected by chance. Here, the *cal_mantel_corr* function in the microeco package was used (93).

To characterize if alpha diversity (i.e., phylogenetic structure) of phytoplankton was governed by environmental filtering or overdispersion we used the NRI index (nearest related index, see also Table 1). For this, first, the phylogenetic metric MPD (mean pairwise distance) was calculated as it considers the mean phylogenetic distance among all pairs of species within a community (44). Phylogenetic structure (alpha diversity) was then assessed by the NRI for each community with null models based on 999 randomizations with the random shuffling of the phylogenetic tree labels with MicrobiotaProcess package v.1.5.4.990 (i.e., the *get_NRI_NTI* function). The NRI index was obtained by multiplying the standardized effect size of the mean pairwise phylogenetic distance by −1.

The phylogenetic temporal turnover between pairwise communities among sampling dates (beta diversity) was quantified to investigate the action of deterministic and stochastic ecological processes with microeco R package v.0.6.0 (93), using the *trans_nullmodel* function. The phylogenetic distance

between pairwise communities (beta mean pairwise distance [$\beta$MPD]) was computed with null models based on 999 randomizations with the random shuffling of the phylogenetic tree labels as in Stegen et al. (34). The $\beta$NRI was calculated via the z score, as the difference between the observed $\beta$MPD and the mean of the $\beta$MPD null models divided by the standard deviation of the null models. $\beta$NRI scores less than $-2$ indicate that the observed phylogenetic turnover was significantly lower than $\sim$95% of the null values and thus that homogeneous selection between the compared communities causes higher than expected phylogenetic similarity. Similarly, $\beta$NRI scores greater than $+2$ indicate the dominance of heterogeneous selection. The Raup-Crick distances based on the Bray-Curtis similarity (RC$_{bray}$) were also calculated to further differentiate the stochastic processes structuring the community assembly when $\beta$NRI scores varied between $-2$ and $+2$ (34, 94). Values, of RC$_{bray}$ less than $-0.95$ or greater than 0.95 indicate less and more compositional turnover, respectively, than the null expectation and that was attributed to homogeneous dispersal in the former case and dispersal limitation in the latter. All definitions of the different assembly processes corresponding to the values of NRI and $\beta$NRI were shown in Table 1. Nonweighted metrics were used as metabarcoding data were semiquantitative and the rarefed data set was considered to prevent any bias due to potential under-sampling (22). The analysis in August has not been considered for further discussion in the present study because of insufficient sampling (i.e., only three samples).

To evaluate whether a phylogenetic structure (i.e., NRI) and turnover (i.e., $\beta$NRI) were attributed to different environmental variables a permutational multivariate analysis of variance (PERMANOVA) was performed based on the Bray-Curtis distance.

**Data availability.** Raw sequencing data have been submitted to the Short Read Archive under BioProject number PRJNA851611.

## SUPPLEMENTAL MATERIAL

Supplemental material is available online only.

**SUPPLEMENTAL FILE 1**, PDF file, 1.3 MB.

## ACKNOWLEDGMENTS

We thank the captain and the crew of the RV 'Sepia II'; M. Crouvoisier for nutrient analysis; V. Cornille and E. Lecuyer for help with the fieldwork; E. Goberville for meteorological data; and P. Magee for English proofing. We thank the SCoSI/ULCO (Service COmmun du Système d'Information de l'Université du Littoral Côte d'Opale) for providing us with the computational resources to run all the bioinformatic analyses via the CALCULCO computing platform (https://www-calculco.univ-littoral.fr/). We also thank the reviewers for their comments and suggestions that helped to improve our manuscript.

This work was logistically supported by the national monitoring network SOMLIT (https://www.somlit.fr/) and funded by the CPER MARCO (https://marco.univ-littoral.fr/) and the French Research program of INSU-CNRS via the LEFE-EC2CO 'PLANKTON PARTY' and the OFB through the INDIGENE project. DIS was funded via a PhD grant by the 'Region des Hauts de France' and the 'Pôle métropolitain de la Côte d'Opale (PMCO)'.

We declare that all authors of the manuscript do not have any conflict of interest.

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
