## [Reviewer comments · Microbiology Spectrum]

Microbiology Spectrum

Stochastic and deterministic processes regulate phytoplankton assemblages in a temperate coastal ecosystem

Dimitra-Ioli Skouroliaou, Elsa Breton, Solène Irion, Luis Artigas, and Urania Christaki

Corresponding Author(s): Dimitra-Ioli Skouroliaou, Laboratory of Oceanography and Geosciences UMR 8187 CNRS ULCO

Review Timeline:

Submission Date:	June 27, 2022
Editorial Decision:	July 26, 2022
Revision Received:	September 16, 2022
Accepted:	September 20, 2022

Editor: Konstantinos Kormas

Reviewer(s): The reviewers have opted to remain anonymous.

Transaction Report:

DOI: <https://doi.org/10.1128/spectrum.02427-22>

July 26, 2022

Mx. Dimitra-Ioli Skouropoliakou
Laboratory of Oceanography and Geosciences UMR 8187 CNRS ULCO
28 Av. du Maréchal Foch
Wimereux 62930
France

Re: Spectrum02427-22 (Ecological drift and homogeneous selection regulate phytoplankton assemblages in a temperate coastal ecosystem)

Dear Mx. Dimitra-Ioli Skouropoliakou:

Reviewer 1 raised the issue of showing the novelty of your research, and this is a desirable aspect in every scientific paper, but SPECTRUM MICROBIOLOGY does not have this as a prerequisite for accepting a paper for publication. Instead, scientific soundness of the submitted is the key factor and for this I'd like to draw your attention to work mostly on the comments related to these aspects.

Link Not Available

Sincerely,

Konstantinos Kormas

Journals Department
Reviewer comments:

Reviewer #1 (Comments for the Author):

Ecological drift and homogeneous selection regulate phytoplankton assemblages in a temperate coastal ecosystem
By Dimitra-Ioli Skouropoliakou, Elsa Breton, Solène Irion, Luis Felipe Artigas, and Urania Christaki

General comments:

This was an interesting study presenting a thorough analysis that assessed the relative role of environmental selection, dispersal and drift as phytoplankton assembly processes. The analysis was based on ample phytoplankton molecular, morphological and environmental data collected with appropriate methods and sample size was substantial covering a 5 year period albeit from a small spatial scale. Although I am not familiar with the methodological framework used in data analysis, the authors seem to have used existing frameworks and applied appropriate null models. I am happy with the quality of results and figures as well as main findings.

However, the manuscript suffers from lack of focus and clarity in the presentation of methods, results and discussion. The main reason behind this is the lack of a clear research question and hypothesis that should be specifically stated for the system in question (given the small spatial scale). Albeit the system is indeed of high interest given the annual emergence of *P.globosa* blooms and the high seasonal turnover in env conditions and phytoplankton. Instead, much of the manuscript contains descriptive sections that are not directly related to the main objective and which dilute the main message and potential novelty of the research. These sections (eg L308-337) should either be omitted altogether, or be placed in the supplementary material and be referred in the main text in a manner that helps formulate a research question and in interpreting the main results. The methods and results thus need to be thoroughly revised in a manner that better links to the question/s set. Any analysis and results presented need to be justified as to the manner they contribute in addressing the question. Descriptive stats that have no actual purpose should not be presented in the main text but rather placed within supplementary material and referred to in results and discussion.

This lack of focus is seen also clearly in the abstract where a clear research question is never clearly articulated. For instance, it would be good to explain what is it that you are hypothesizing for the particular system, which process will prevail? This lack of clarity is not helped by the use of jargon terminology in the abstract that is not explained although fundamental to the research question. These terms (eg drift vs selection) should be mentioned and briefly clarified first, before paying down the question and hypothesis. The results from the different methodological approaches should again be better linked to the question being addressed (eg which indicates drift and which selection and was their relative importance assessed). The abstract also fails to transmit what is the main novel finding from this study.

I believe the manuscript should not be considered for publication unless this deep revision process takes place to bring out the potential novelty of the findings. I was not able to assess the novelty at this point given the convoluted structure, lack of hypothesis and focus on descriptive rather than the interesting and directly relevant findings.

Specific comments:

L8: I guess such processes are not always opposed but can also act in synergy. I would delete opposed from here as so far, there has been no context provided to justify such a statement.

L8-14: I would have expected a clear research question and hypothesis to be presented somewhere at this point, before embarking on more methodological aspects. Exploring patterns sounds too general and vague and does not persuade as to the novel points that the study is addressing...Eg, it is unclear why the specific methodological approaches were used when the actual question has not been presented yet.

L14: unclear what "to be qualified" means in this context..

L17: "drift and by homogeneous selection" are terms that cannot be expected to be understandable by all readers, of the same or different disciplines.

L32-33: Syntax is off here, please rephrase

L34-35: It is unclear what the authors claim to be the novelty here. Surely there have been studies in the past investigating phytoplankton assembly processes using even longer timeseries (see eg: <https://www.nature.com/articles/s41598-022-07009-6>) or the relative role of dispersal vs environment in both phytoplankton and bacterial communities (<https://www.ncbi.nlm.nih.gov/pmc/articles/PMC7540538/>, <https://pubmed.ncbi.nlm.nih.gov/31662088/>). The authors need to be very clear what exactly is the gap and how their own study is addressing a novel question/approach/finding.

Lines 54-62: reading these lines was very refreshing as this part of the ms is very clear, flows well. However, I would like the authors to make an effort to better identify the gap: they mention that microbial ecologists have lately focused on the relative importance of stochastic vs deterministic process - but not in marine phytoplankton? First of all, studies that have addressed this in marine phytoplankton are missing and they should not. Second, you will need to state why you expect that marine phytoplankton will follow different paradigms to the ones revealed in terrestrial and freshwater assemblages - why would the rules change in the marine environment? There is plenty to be said here to bring out the gap..

L66-73. This is a good description of the assemblage and env however it does not justify why it is an interesting system to investigate these processes. You really need to be more specific here.

L76-77. Syntax needs correcting here.

L78-80. I find the stating of this question a bit confusing as the main processes have just been described and was specifically stated that it was not the environment but rather inter-taxa relations and environmental filtering including competition and predation. Thus what exactly is the gap that the authors are wanting to fill here. Where there any limitations in the previous studies and how is your approach helping take knowledge in this field a step forward?

L81-89. No clear research question and hypothesis specific to the system in question.

Methods

L152-154: Fair enough, but you need to justify in which the use of these methods is helping you address your specific research question? If it was to describe the system then, explicitly say so. Also here you are mentioning quite different methods, from alpha and beta diversity indices to non parametric statistical tests. Why did you pick the specific ones?

L156: what does "it" refer to? Please rephrase.

L157-161. I find this brief explanation very useful. However, as I am unfamiliar with this approach, it was unclear to me at this point (before having read the results) whether the tests described afterwards, are part of Stegen's approach eg Mentel tests, MPD..

L161-162. Provide an example to illustrate this better. Eg when phylogenetic turnover is high this translates as...

L166. Significant correlations with what. It is unclear what is being correlated with what here.

L166. You need to justify the use of both molecular and metabarcoding data. Do you use both for validation purposes or to get complementary information? The only point that this becomes clear, is in the first sentence of the discussion, while it should come much sooner.

L170: it is best to say in the beginning of this paragraph for what purpose you used the NRI index for. Generally, I really appreciate you kept the methods short without giving too much technical details while pointing to supplementary information and further sources for more details. However, when you provide more details on the different indices and data used, you really need to connect them better with how they help you obtain an indication of the relative role of selection, drift, dispersal. This was not clear from the data analysis section.

L194-215. Are these patterns expected and do they provide an indication on what we should be expecting regarding the dominant processes acting on the phytoplankton assemblage? Personally I am not a great fan of descriptive stats and graphs when these do not directly link either to a hypothesis or data interpretation..

L216-266. Again I am not seeing how this is linked to something...All this is extremely descriptive and needs to be summarised in a way that it links with the main results further down. At present these sections are too disconnected and it is hard to understand their point. Perhaps restructuring might help...eg present this info after you have shown the role of selection to explain it further? Or else summarise appropriately and guide the readers better as to how this result is linked/helps address your question or expected findings.

L269-283: In each of these paragraphs, please remind the readers what is the physical interpretation of each result ie what does this positive correlation indicate and which aspect of the question it helps address.

Discussion:

L308: Before going into less related matters, I would see more discussion on the main findings presented in the first paragraph. I would like to see these discussed based on an initial hypothesis regarding the system. I would also like to see these discussed based on previous studies investigating the relative importance of similar processes. Best to move directly to line 338.

L309-337: Very descriptive info and I don't see how it is related to the research question unless some of this info is carefully integrated into the important paragraphs to help elucidate the processes. I would delete these paragraphs altogether as they only draw away from your message.

L348-350: Try to use less jargon and instead interpret the findings making appropriate links with the specific environmental pressures the system is experiencing.

L353-354: why "likely"? Isn't this something that you could test with your dataset?

L364-366: This result seems to contradict previous experimental observations on the importance of drift increasing under high selection and low dispersal (<https://www.nature.com/articles/s41396-020-00754-4>). It would be good to see how the authors

comment on this.

L393: Unclear what you mean by "out-passed" here..

Reviewer #2 (Comments for the Author):

This is an elegant work, and a well-written manuscript tackling a long-posed question: Is phytoplankton succession influenced by deterministic processes (e.g. environmental filtering and biotic interactions) or by randomness (e.g. dispersal, ecological drift etc)? Using a multifaceted analysis and the eastern English Channel as a model system, the authors examined a 5-year dataset of frequent samplings over a small spatial scale of 5 sampling points within 15 km. They suggest that seasonal dynamics of phytoplankton assemblages in most of the year in the sampling area can be unpredictable as ecological drift seems to overcome selection. I really enjoyed the paper. The methods are to the best of my knowledge scientifically sound, and the statistical tools used appropriate. I have some minor comments and proposed amendments below:

Abstract:

The abstract seemed a bit complicated and difficult to follow. For example, L27: neutral-dominated community, L17 and title: ecological drift and homogeneous selection, L19: deterministic homogeneous selection, L20: stochastic ecological drift, L16: environmental filtering, L23: stochastic processes. All these terms in my understanding essentially refer to stochastic and deterministic processes. The use of all these different terms for the same general process is a bit confusing. In the introduction and Table 1 everything is clearer, but the word count here is limiting to elaborate, so you could cluster these terms referring to a process under one term. Also, please rephrase L13-14: "Microscopy and flow-cytometry quantified phytoplankton biomass to be qualified", what do you mean by "to be qualified"?

Results:

Tables need a bit of editing. E.g. central alignment seems not accurate in some columns.

Table 2: Give full genera names in abbreviated taxa of dominant phytoplankton. Same in-text when mentioning the species for the first time (e.g. L249 *C. socialis*).

Figure 1 is very nice and informative. However, I wonder why use both statistical tools. The outcome is essentially the same, with db-RDA including and expanding the information of PCA as far as I can understand.

L37-42 in Supplementary material and Figure S10. I am a bit confused by the methods and what the figure represents. Can you elaborate a bit here? This was done for all ASVs pairwise? And what each square represents in the Figure?

Discussion

L339-347: I am a bit confused here: In the beginning of the paragraph the authors discuss that environmental filtering, a deterministic process, is evidenced for phytoplankton community assembly and dynamics. But then they seem to contradict themselves by mentioning that drift was more important than selection during most of the year. Please clarify.

L370-373: I think this is redundant, all these mechanisms could be considered deterministic rather than random. For example, productivity depends on community structure and abundances, which can be driven by environmental filtering as shown.

L403: Is *P. pungens* a mixotroph? According to the listed references or my knowledge, I couldn't confirm this, although in Burkholder et al it is mentioned that *P. australis* can employ osmotrophy.

L441-443 are repeating the same text as few lines above. You could delete L427-428 and change accordingly the text.

Staff Comments:

Preparing Revision Guidelines

Please return the manuscript within 60 days; if you cannot complete the modification within this time period, please contact me. If you do not wish to modify the manuscript and prefer to submit it to another journal, please notify me of your decision immediately so that the manuscript may be formally withdrawn from consideration by Microbiology Spectrum.

Response to the reviewers

Reviewer comments:

Reviewer #1 (Comments for the Author):

Ecological drift and homogeneous selection regulate phytoplankton assemblages in a temperate coastal ecosystem

By Dimitra-Ioli Skouroliakou, Elsa Breton, Solène Irion, Luis Felipe Artigas, and Urania Christaki

General comments:

This was an interesting study presenting a thorough analysis that assessed the relative role of environmental selection, dispersal and drift as phytoplankton assembly processes. The analysis was based on ample phytoplankton molecular, morphological and environmental data collected with appropriate methods and sample size was substantial covering a 5-year period albeit from a small spatial scale. Although I am not familiar with the methodological framework used in data analysis, the authors seem to have used existing frameworks and applied appropriate null models. I am happy with the quality of results and figures as well as main findings.

However, the manuscript suffers from lack of focus and clarity in the presentation of methods, results and discussion. The main reason behind this is the lack of a clear research question and hypothesis that should be specifically stated for the system in question (given the small spatial scale). Albeit the system is indeed of high interest given the annual emergence of *P.globosa* blooms and the high seasonal turnover in env conditions and phytoplankton. Instead, much of the manuscript contains descriptive sections that are not directly related to the main objective and which dilute the main message and potential novelty of the research.

These sections (eg L308-337) should either be omitted altogether, or be placed in the supplementary material and be referred in the main text in a manner that helps formulate a research question and in interpreting the main results.

-The methods and results thus need to be thoroughly revised in a manner that better links to the question/s set.

Any analysis and results presented need to be justified as to the manner they contribute in addressing the question.

Descriptive stats that have no actual purpose should not be presented in the main text but rather placed within supplementary material and referred to in results and discussion.

-This lack of focus is seen also clearly in the abstract where a clear research question is never clearly articulated.

For instance, it would be good to explain what is it that you are hypothesizing for the particular system, which process will prevail?

-This lack of clarity is not helped by the use of jargon terminology in the abstract that is not explained although fundamental to the research question. These terms (eg drift vs selection) should be mentioned and briefly clarified first, before paying down the question and hypothesis.

-The results from the different methodological approaches should again be better linked to the question being addressed (eg which indicates drift and which selection and was their relative importance assessed). The abstract also fails to transmit what is the main novel finding from this study.

I believe the manuscript should not be considered for publication unless this deep revision process takes place to bring out the potential novelty of the findings. I was not able to assess the novelty at this point given the convoluted structure, lack of hypothesis and focus on descriptive rather than the interesting and directly relevant findings.

Response Reviewer 1 General comments

The major concerns - repeated many times and in different ways throughout the review- were synthesized below.

The reviewer:

1. asks to bring out the potential novelty of the study and to formulate a clear hypothesis. This comment is also found in four specific comments as 'gap'

We thank the reviewer for this comment which help to put forward our work.

In the revised version of the manuscript several modifications were made to better bring out the novelty ('gap') and a clear hypothesis was formulated.

changes made:

1. The abstract was extensively rewritten to make it more accessible (reviewer 2). The hypothesis is formulated (**L10-L12**).

It now reads: *'In this study we hypothesized that deterministic and stochastic ecological processes regulating phytoplankton, present seasonal and repeating patterns.'*

2. In the 'Importance', the novelty of the study was addressed, but it is now rephrased to explicitly put it forward (**L30-L35**, see also reviewer 2).

It now reads: *'Understanding the overall assembly processes of phytoplankton is critical in tracking and predicting future changes. The novelty of this study is that it addresses for the first time, and on a pluri-annual scale, a long-posed question: Is seasonal phytoplankton succession influenced by deterministic processes (e.g., abiotic environment) or by stochastic ones (e.g., dispersal, or ecological drift). Our results provided strong support for a seasonal and repeating pattern with stochastic processes (drift) prevailing during most of the year, and in particular, during the periods that presented transient monospecific phytoplankton peaks.'*

3. In the introduction a paragraph was added to distinguish our study relative to previous ones (**L62-L72**).

As follows: *'Historically, phytoplankton assemblages have been studied from a deterministic perspective based on their traits (e.g., 13, 14), and their environment (15). However, deterministic processes in structuring phytoplankton communities (16) are insufficient to explain overall community structure and diversity patterns (17). Marine phytoplankton studies have been partly focused on stochastic (e.g., 18) or dispersal processes (e.g., 19). Yet, there is a need to understand how deterministic and stochastic processes potentially change in one ecosystem at different time scales (11, 20). Two existing phytoplankton studies have quantified the relative contribution of both deterministic and stochastic processes focused on large spatial scale (21) or short time periods (less than a year) (22). However, the present study is the first one to quantify both deterministic and stochastic phytoplankton assembly processes at a seasonal scale over a pluri-annual sampling period.'*

4. The hypothesis is also formulated in the introduction (**L87-89**).

As follows: *'We hypothesized that deterministic and stochastic ecological processes regulating phytoplankton, present seasonal and repeating patterns.'*

2. considers that the abstract contains too much of jargon terminology and lacks of a clear question

We agree, following also the suggestions of reviewer 2 the title and the abstract were modified to make them more accessible.

All changes are highlighted in blue to felicitate reading.

Table 1 was also slightly modified (see below)

3a. reviewer requires to delete from the results and discussion the description of the community structure and succession

We are perplexed with these requirements. Although actually some authors present exclusively statistical analysis of communities without describing them, many others think that the opposite is right (please also see the paper that you suggested by Aguilar and Sommaruga 2019, cited in our paper). What the reviewer characterizes as '*not related to the main objective*', '*having no actual purpose*' etc ... is the core of our work. It is necessary to present the community description and more importantly the succession, as a background to the presentation of the ecological processes witnessed here. These results are the core of the ecological processes analysis which could have not existed without them. There is no scientific argument to delete the description of the communities which we consider very important to make a sound paper.

3b. is not 'fun' of descriptive statistics which have no 'actual purpose'

As in the previous paragraph, descriptive statistics are necessary as a first step to summarize tendencies before realizing more elaborate numerical analysis. For example, when we say that richness is higher in autumn, or that nutrients are associated with so and so, our readers need to have some evidence as to where this information comes from.

Besides, Figure 1 (PCA, db-RDA) are not 'descriptive' but multivariate statistics and are necessary to summarize and illustrate the degree of seasonal variation we observed during these 5 years. The box plots summarize a big amount of data in order to help to follow the community structure and succession (see previous paragraph).

We hope that the reviewer can hear our arguments.

Specific comments:

L8: I guess such processes are not always opposed but can also act in synergy. I would delete opposed from here as so far, there has been no context provided to justify such as statement.

Absolutely! 'opposed' is deleted

L8-14: I would have expected a clear research question and hypothesis to be presented somewhere at this point, before embarking on more methodological aspects. Exploring patterns sounds too general and vague and does not persuade as to the novel points that the study is addressing...Eg, it is unclear why the specific methodological approaches were used when the actual question has not been presented yet.

see above

The abstract was extensively rewritten, the research question and hypothesis was added. All changes are highlighted in blue to facilitate reading (L9-L12).

It now reads: '*...understanding and predicting community organization and succession at different temporal and spatial scales. In this study we hypothesized that deterministic and stochastic ecological processes regulating phytoplankton, present seasonal and repeating patterns.*'

L14: unclear what "to be qualified" means in this context.

Sorry this was a typo, changed into 'quantified' (L14).

L17: "drift and by homogeneous selection" are terms that cannot be expected to be understandable by all readers, of the same or different disciplines.

Thank you for this comment, we agree, following also the suggestions of reviewer 2 the title and the abstract were modified to make them more accessible. Table 1 has been slightly modified (see below).

L32-33: Syntax is off here, please rephrase

It was simplified into: '*Understanding the overall assembly processes of phytoplankton is critical to tracking and predicting future changes*' (L30-L31).

L34-35: It is unclear what the authors claim to be the novelty here. Surely there have been studies in the past investigating phytoplankton assembly processes using even longer timeseries (see eg: <https://www.nature.com/articles/s41598-022-07009-6>) or the relative role of dispersal vs environment in both phytoplankton and bacterial communities (<https://www.ncbi.nlm.nih.gov/pmc/articles/PMC7540538/>,

<https://pubmed.ncbi.nlm.nih.gov/31662088/>). The authors need to be very clear what exactly is the gap and how their own study is addressing a novel question/approach/finding.

In the revised version of the manuscript several modifications were made to better bring out the novelty ('gap') and a clear hypothesis was formulated.

Detailed in **point 1** above and **next comment**. The references suggested by the reviewer were added and discussed relative to the present study **L55, L63, L66**.

References:

- Aguilar P, Sommaruga R. 2020. The balance between deterministic and stochastic processes in structuring lake bacterioplankton community over time. *Molecular Ecology* 29:3117–3130.
- Longobardi L, Dubroca L, Margiotta F, Sarno D, Zingone A. 2022. Photoperiod-driven rhythms reveal multi-decadal stability of phytoplankton communities in a highly fluctuating coastal environment. 1. *Sci Rep* 12:3908.
- Spatharis S, Lamprinou V, Meziti A, Kormas KA, Danielidis DD, Smeti E, Roelke DL, Mancy R, Tsirtsis G. 2019. Everything is not everywhere: can marine compartments shape phytoplankton assemblages? *Proc Biol Sci* 286:20191890.

Lines 54-62: reading these lines was very refreshing as this part of the ms is very clear, flows well. However, I would like the authors to make an effort to better identify the gap: they mention that microbial ecologists have lately focused on the relative importance of stochastic vs determinist process - but not in marine phytoplankton? First of all, studies that have addressed this in marine phytoplankton are missing and they should not. Second, you will need to state why you expect that marine phytoplankton will follow different paradigms to the ones revealed in terrestrial and freshwater assemblages - why would the rules change in the marine environment? There is plenty to be said here to bring out the gap.

a) This comment joins the point 1 and the previous comment.

These issues are the same as point 1 and the previous comment and are detailed above

The 'gap', 'novelty' and hypothesis are now added

More references on phytoplankton were added.

b) We never said that we expected that marine phytoplankton will follow different paradigms to the ones revealed in terrestrial and freshwater assemblages. We just said that this approach has been applied to these environments. Thus, we do not necessarily expect that marine phytoplankton will follow different paradigms than the terrestrial or freshwater ones.

To further clarify this issue, we added the following sentences (**L54-L55** and **L60-L61**).

L54-L55: *'...by using concepts developed in terrestrial ecology'*

L60-L61: *'The majority of literature developing ecological theories (such as, Hubbel's, 2001) and the respective methodology derive from terrestrial ecology.'*

c) As already mentioned, we also added few sentences positioning our study relative to previous ones (**L65-L72**).

It now reads: *'Marine phytoplankton studies have been partly focused on stochastic (e.g., 18) or dispersal processes (e.g., 19). Yet, there is a need to understand how deterministic and stochastic processes potentially change in one ecosystem at different time scales (11, 20). Two existing phytoplankton studies have quantified the relative contribution of both deterministic and stochastic processes focused on large spatial scale (21) or short time periods (less than a year) (22). However, the present study is the first one to quantify both deterministic and stochastic phytoplankton assembly processes at a seasonal scale over a pluri-annual sampling period.'*

L66-73. This is a good description of the assemblage and env however it does not justify why it is an interesting system to investigate these processes. You really need to be more specific here.

The reviewer recognizes in the introductory statement (see page 1) that *'Albeit the system is indeed of high interest given the annual emergence of P.globosa blooms and the high seasonal turnover in env conditions and phytoplankton'*

This paragraph was modified using the reviewer's statement. (**L73-L74**)

It now reads: *'Given the annual emergence of P. globosa blooms and the high seasonal turnover in environmental conditions and phytoplankton...'*

L76-77. Syntax needs correcting here.

It now reads: *'Previous studies investigating drivers structuring planktonic communities in the eastern English Channel focused on deterministic processes such as inter-taxa relations (31), environmental filtering (32), and predation (33).'* (**L83-L85**)

L78-80. I find the stating of this question a bit confusing as the main processes have just been described and was specifically stated that it was not the environment but rather inter-taxa relations and environmental filtering including competition and predation. Thus, what exactly

is the gap that the authors are wanting to fill here. Where there any limitations in the previous studies and how is your approach helping take knowledge in this field a step forward?

The overall novelty and 'gap' were described above, as consequence this sentence was deleted.

We only left a sentence:

'The present study aimed to explore the stochastic and deterministic ecological processes driving the seasonal organisation of phytoplankton assemblages, and how these processes varied across seasons. (L85-L87)

L81-89. No clear research question and hypothesis specific to the system in question.

The research question was stated in the submitted version and is also now stated in lines L31-L34 and L85-L87.

It now reads:

L31-L34: *'The novelty of this study is that it addresses for the first time, and on a pluri-annual scale, a long-posed question: Is seasonal phytoplankton succession influenced by deterministic processes (e.g., abiotic environment) or by stochastic ones (e.g., dispersal, or ecological drift).'*

L85-L87: *'The present study aimed to explore the stochastic and deterministic ecological processes driving the seasonal organisation of phytoplankton assemblages, and how these processes varied across seasons.'*

The hypothesis has been now added here (**L87-89**)

'We hypothesized that deterministic and stochastic ecological processes regulating phytoplankton, present seasonal and repeating patterns.'

Methods

L152-154: Fair enough, but you need to justify in which the use of these methods is helping you address your specific research question? If it was to describe the system then, explicitly say so. Also, here you are mentioning quite different methods, from alpha and beta diversity indices to non-parametric statistical tests. Why did you pick the specific ones?

The main question in this study is whether the stochastic and deterministic ecological processes varied across seasons. Thus, first we had to confirm the seasonal variations of the environmental variables (PCA), and the seasonal variation of the phytoplankton communities based on the environmental variables (db-RDA).

The methods used are adequate.

As we developed in point in point 3b of general comments the description of the environment is a necessary step to summarize tendencies before realizing the elaborate numerical analysis used here. Alpha, beta diversity and comparison between stations etc are part of this.

We think that this was not needed to be explained further. Although we wished to keep this section short and concise and not to stuff the paper with details, to please the reviewer, we added why this was done and more details about each test.

It now reads:

L159-L164: *'To describe the phytoplankton communities, the alpha diversity and Faith's phylogenetic diversity indices were calculated. Kruskal-Wallis and the post-hoc Nemenyi test were used to test if phytoplankton groups significantly differ among stations. To explore and summarize seasonal variations in the abiotic environment the multivariate Principal Component Analysis (PCA) was performed. Distance-based RDA analysis was applied considering morphological and metabarcoding data. All analysis was performed in R (46).'*

Below we note several parts where these analyses were used in the discussion section to help the reviewer to realize why they were necessary.

Alpha diversity indices, multivariate, and PERMANOVA analyses allowed as a better understanding of the processes that prevailed in different seasons.' (e.g., in discussion section)

L419-L422: *'This was coherent with the large quantity of unexplained variance between environmental variables and the phytoplankton community in terms of abundance, and phylogenetic structure and turnover (db-RDA and PERMANOVA, respectively).'*

L450 – L453: *'Despite the relatively low nutrient levels after the P. globosa bloom, phytoplankton richness and PD values increased (Table 2) and several transient peaks belonging to different phylogenetic groups (diatoms, dinoflagellates, and picocyanophytoplankton) appeared in summer (Fig. 4) (27).'*

L156: what does "it" refer to? Please rephrase.

Changed into: *'Briefly, the community assembly analysis...'* **L168**

L157-161. I find this brief explanation very useful. However, as I am unfamiliar with this approach, it was unclear to me at this point (before having read the results) whether the tests described afterwards, are part of Stegen's approach eg Mentel tests, MPD..

An additional explicative phrase was added: *'All metrics and tests were also applied according to this framework.'* **L167**

L161-162. Provide an example to illustrate this better. Eg when phylogenetic turnover is high this translates as...

This comment did not refer to L161-162 but L156-161 of the initial submitted version.

We agree that this could be made more accessible.

This paragraph was rewritten with more explanations as follows:

L172-L184: *'According to phylogenetic community composition (alpha diversity) within each sample, phylogenetic structure is divided into environmental filtering and overdispersion, which are deterministic processes (Table 1). Based on phylogenetic turnover in community composition (beta diversity), deterministic processes are divided into homogeneous selection (i.e., consistent environmental factors cause low compositional turnover) and heterogeneous selection (i.e., high compositional turnover caused by shifts in environmental factors). However, stochastic processes are divided into homogeneous dispersal (i.e., low compositional turnover caused by high dispersal rates), dispersal limitation (i.e., high compositional turnover caused by a low rate of dispersal), and ecological drift that can result from fluctuations in population sizes due to chance events (7, 34, 47, Table 1). The implicit hypothesis is that phylogenetic conservatism exists, which means that ecological similarity between taxa is related to their phylogenetic similarity (i.e., phylogenetic signal).'*

Also note that these modifications are included in table 1

L166. Significant correlations with what. It is unclear what is being correlated with what here.

Clarified as follows: **L184-L186**

'Mantel correlograms were applied to detect phylogenetic signal, correlating the phylogenetic distances to the niche distances at different distance classes'

L166. You need to justify the use of both molecular and metabarcoding data. Do you use both for validation purposes or to get complementary information? The only point that this becomes clear, is in the first sentence of the discussion, while it should come much sooner.

We agree with the reviewer and this is why in the abstract it was made clear in the submitted and revised version what we expected from each method *' Microscopy and flow-cytometry quantified phytoplankton abundance and biomass while metabarcoding data allowed an extended evaluation of diversity and the exploration of the ecological processes regulating phytoplankton, using null model analysis'* (**L14-L17**)

We do not think that we should repeat this sentence again and L166 was not the right place to add this

In the beginning of the discussion it was also stated (submitted version) that:

L345-L346: *'In the present study, microscopy and flow cytometry allowed phytoplankton biomass to be quantified, while metabarcoding data provided an extended evaluation of its diversity'*

In the current version, we also state the importance of good correspondence between morphological and metabarcoding data as follows:

L499-L501: *'Molecular and morphological data are complementary but unfortunately are rarely considered together in actual marine planktonic studies, and this is one of the strong points of our work.'*

We do not think it is useful to add more and lengthen the paper unnecessarily.

To please however the reviewer the following was added in results:

L257-L263: *'Our study is one of the rare ones considering both morphological and metabarcoding data using a relatively large data set (287 samples). Accordingly, since metabarcoding data are subjected to PCR biases and are always expressed in relative abundances, while morphological data are absolute abundances, it was important to confront the two data sets and see if they show similar trends. For this, the db-RDA was also applied to the microscopy data, which revealed similar seasonal trends (Fig. S3).'*

For the reviewer

This is also what we discussed here in the paragraph 'seasonal patterns'. (see also answer to the comment where the reviewer asked to delete this section). We also discussed why we decided to use the data of the two small dinoflagellates that increase in summer, after verification that they were not over-represented in metabarcoding data

The reviewer can appreciate also that this synthesis of morphological and metabarcoding data was realized with a substantial data set and this can be useful for future studies.

L170: it is best to say in the beginning of this paragraph for what purpose you used the NRI index for. Generally, I really appreciate you kept the methods short without giving too much technical details while pointing to supplementary information and further sources for more details. However, when you provide more details on the different indices and data used, you

really need to connect them better with how they help you obtain an indication of the relative role of selection, drift, dispersal. This was not clear from the data analysis section.

Thank you, we wished to keep this part short

Explicative sentences were added before providing more details on the different methods.

-For alpha diversity (i.e., phylogenetic structure):

L191-L193: *'To characterise if alpha diversity (i.e., phylogenetic structure) of phytoplankton is governed by environmental filtering or overdispersion we used the NRI index (Nearest Related Index, see also Table 1).'*

- For beta diversity (i.e., phylogenetic temporal turnover):

L207-L216: *' β NRI scores less than -2 indicate that the observed phylogenetic turnover is significantly lower than ~95% of the null values and thus that homogeneous selection between the compared communities causes higher than expected phylogenetic similarity. Similarly, β NRI scores greater than +2 indicate the dominance of heterogeneous selection. The Raup-Crick distances based on the Bray-Curtis similarity (RC_{bray}) were also calculated to further differentiate the stochastic processes structuring the community assembly, when β NRI scores varied between -2 and +2 (34, 52). Values, of RC_{bray} less than -0.95 and greater than 0.95 indicate less and more compositional turnover, respectively, than the null expectation and that is attributed to homogeneous dispersal in the former case and to dispersal limitation in the latter.'*

Results:

L194-215. Are these patterns expected and do they provide an indication on what we should be expecting regarding the dominant processes acting on the phytoplankton assemblage?

Personally, I am not a great fan of descriptive stats and graphs when these do not directly link either to a hypothesis or data interpretation.

This has been answered many times before and in general comments 3a and 3b.

We need the description of the seasonal description of abiotic and biotic variables prior to 'processes' analysis

L216-266. Again, I am not seeing how this is linked to something...All this is extremely descriptive and needs to be summarised in a way that it links with the main results further down. At present these sections are too disconnected and it is hard to understand their point.

Perhaps restructuring might help...eg present this info after you have shown the role of selection to explain it further? Or else summarise appropriately and guide the readers better as to how this result is linked/helps address your question or expected findings.

This also joins general comment 3a and 3b, our answer detailed above. We do not see what the reviewer wants to 'reconstruct' here. Clarifications for the use of db-RDA for morphological data are now added. It reads as follows:

L257-L262: 'Our study is one of the rare ones considering both morphological and metabarcoding data using a relatively large data set (287 samples). Accordingly, since metabarcoding data are subjected to PCR biases and are always expressed in relative abundances, while morphological data are absolute abundances, it was important to confront the two data sets and see if they show similar trends. For this...'

L269-283: In each of these paragraphs, please remind the readers what is the physical interpretation of each result ie what does this positive correlation indicate and which aspect of the question it helps address.

Explicative sentences were added in the text as follows:

L309-310: '*...correlating the phylogenetic distances to the niche distances at different distance classes detected significant positive correlations across short phylogenetic distances...*' and

L313-L314: '*These results supported that phylogenetic metrics can be applied to infer ecological assembly processes.*'

L316-L318: '*Ecological processes governed the alpha diversity of phytoplankton (i.e., phylogenetic structure) were investigated using the NRI index. It showed that, phylogenetic clustering prevailed during all seasons suggesting environmental filtering...*'

L323-L324: '*To test if phylogenetic structure (i.e., NRI) and turnover (i.e., β NRI) were attributed to different environmental variables a PERMANOVA was applied.*'

L330: '*The different ecological processes, selection, dispersal and drift were quantified using...*'

Discussion:

L308: Before going into less related matters, I would see more discussion on the main findings presented in the first paragraph. I would like to see these discussed based on an initial hypothesis regarding the system. I would also like to see these discussed based on previous

studies investigating the relative importance of similar processes. Best to move directly to line 338.

We added the following sentences in the first paragraph as follows:

L346-349: *'Alpha diversity of phytoplankton communities was regulated by environmental filtering. Drift, followed by homogeneous selection, were the major mechanisms regulating the temporal turnover in community composition (beta diversity) and prevailed across seasons.'*

L356-358: *'Overall, we evidenced that deterministic and stochastic ecological processes varied across seasons in alpha and beta diversity.'*

The answer to the hypothesis posed here is explicit in this first paragraph of the discussion.

L349-L358: *'Three periods were evidenced: (i) Winter-early spring, with homogeneous selection as the major process regulating the phytoplankton communities, composed mainly of diatoms communities (e.g., Thalassiosira); (ii) Spring-summer, with drift as the major process in community assembly during the bloom of P. globosa and during the transient peaks of various taxa (diatoms, dinoflagellates, and pico-nanophytoplankton); and (iii) Autumn, with a combination of drift and homogeneous selection as a major ecological process in phytoplankton community assembly dominated by diatoms (Table 2). Overall, we evidenced that deterministic and stochastic ecological processes varied across seasons in alpha and beta diversity.'*

L309-337: Very descriptive info and I don't see how it is related to the research question unless some of this info is carefully integrated into the important paragraphs to help elucidate the processes. I would delete these paragraphs altogether as they only draw away from your message.

Please see our many explanations above. There is no scientific argument to delete the description of the communities which we consider very important to make a sound paper.

L348-350: Try to use less jargon and instead interpret the findings making appropriate links with the specific environmental pressures the system is experiencing.

This is rephrased as follows:

L401-403: *'The dominance of homogeneous selection in winter and early spring is coherent with the strong environmental filtering conditions, such as, the high nutrient concentration values, and low light availability recorded during this period (NRI>4 Fig. 5A, see also Table 1, and 2).'*

Also, to better explain the action of homogeneous selection in the system in winter and early spring we added the following sentence

L408-L410: *'In the geographic scale of our study (ca 15 km), the coastal waters of the eastern English Channel represented an homogeneous pool of phytoplankton taxa undergoing similar selection processes (Figs. 6, S4).'*

L353-354: why "likely"? Isn't this something that you could test with your dataset?

Yes this is something that is confirmed from our dataset, as well as other studies. 'Likely' is now removed. It now reads:

L406-408: *'This process dominates in community assembly when environmental conditions are spatially homogenous (Fig. S2B; 20, 47).'*

L364-366: This result seems to contradict previous experimental observations on the importance of drift increasing under high selection and low dispersal (<https://www.nature.com/articles/s41396-020-00754-4>). It would be good to see how the authors comment on this.

Fodelianakis et al., 2021 used simulations to study drift based on different scenarios of dispersal and selection in synthetic bacterial communities. What he called selection in the paper (which is the 'causal' parameter) is the action of environmental fluctuation. In our study we used null models based on metabarcoding data in natural communities. The major 'causal' parameter of drift was the input of organic material derived from the wane of *Phaeocystis globosa* which can be considered as a natural perturbation enhancing drift (see **L458-L461**). So, their results do not contradict ours, they just used different terminology and method and it not contradicting ours. We do not consider that it is useful to comment this paper in our discussion. However, based on your comment we added a reference to justify our findings as follows: *'The dominance of drift is in accordance with a previous study quantifying the ecological processes in natural ecosystems using the same analytical approaches of the present stud (75) . These authors found that microeukaryotic communities were governed by drift (72%), while the relative contribution of selection and dispersal were low.'* (**L422-L425**)

L393: Unclear what you mean by "out-passed" here..

changed into 'lowered the' **L453**

Reviewer #2 (Comments for the Author):

This is an elegant work, and a well-written manuscript tackling a long-posed question: Is phytoplankton succession influenced by deterministic processes (e.g. environmental filtering and biotic interactions) or by randomness (e.g. dispersal, ecological drift etc)? Using a multifaceted analysis and the eastern English Channel as a model system, the authors examined a 5-year dataset of frequent samplings over a small spatial scale of 5 sampling points within 15 km. They suggest that seasonal dynamics of phytoplankton assemblages in most of the year in the sampling area can be unpredictable as ecological drift seems to overcome selection. I really enjoyed the paper. The methods are to the best of my knowledge scientifically sound, and the statistical tools used appropriate. I have some minor comments and proposed amendments below:

Abstract:

The abstract seemed a bit complicated and difficult to follow. For example,

L27: neutral-dominated community,

L17 and title: ecological drift and homogeneous selection,

L19: deterministic homogeneous selection,

L20: stochastic ecological drift,

L16: environmental filtering,

L23: stochastic processes.

All these terms in my understanding essentially refer to stochastic and deterministic processes. The use of all these different terms for the same general process is a bit confusing. In the introduction and Table 1 everything is clearer, but the word count here is limiting to elaborate, so you could cluster these terms referring to a process under one term. Also, please rephrase

The abstract was extensively rewritten to make it more accessible. **L8-L27**

The title of the article is also modified for the same reason. **L1-L2**

Also, the Table 1 has been slightly modified to better explain the different ecological processes.

All modifications are highlighted in blue to facilitate reading

L13-14: "Microscopy and flow-cytometry quantified phytoplankton biomass to be qualified", what do you mean by "to be qualified"?

Sorry this was a typo, changed into 'quantified' (L14).

Results:

-Tables need a bit of editing. E.g. central alignment seems not accurate in some columns. Central alignment is now corrected. The text of table 1 is also slightly modified, to better explain the different processes.

-Table 2: Give full genera names in abbreviated taxa of dominant phytoplankton. Same in-text when mentioning the species for the first time (e.g. L249 *C. socialis*).

We corrected the Table 2 and the text when mentioning the species for the first time. (L289 and L299).

-Figure 1 is very nice and informative. However, I wonder why use both statistical tools. The outcome is essentially the same, with db-RDA including and expanding the information of PCA as far as I can understand.

The main question in this study is whether the stochastic and deterministic ecological processes varied across seasons. Thus, first we had to confirm the seasonal variations of the environmental variables (PCA), and the seasonal variation of the phytoplankton communities based on the environmental variables (db-RDA) and verify that they show similar trends. An explicative phrase in Material and Methods section to justify the use of these methods, was added as follows:

L161-L164: *'To explore and summarize seasonal variations in the abiotic environment the multivariate Principal Component Analysis (PCA) was performed. Distance-based RDA analysis was applied considering morphological and metabarcoding data. All analysis was performed in R (46).'*

Also, given that metabarcoding data are subjected to PCR biases and are always expressed in relative abundances, while morphological data are absolute abundances, it is important to confront the two data sets and see if they show similar trends, thus db-RDA was applied to both morphological and metabarcoding data. This is stated in the discussion section as follows:

L257-263: *'Our study is one of the rare ones considering both morphological and metabarcoding data using a relatively large data set (287 samples). Accordingly, since metabarcoding data are subjected to PCR biases and are always expressed in relative abundances, while morphological data are absolute abundances, it was important to confront*

the two data sets and see if they show similar trends. For this, the db-RDA was also applied to the microscopy data, which revealed similar seasonal trends (Fig. S3).'

-L37-42 in Supplementary material and Figure S10. I am a bit confused by the methods and what the figure represents. Can you elaborate a bit here? This was done for all ASVs pairwise? And what each square represents in the Figure?

We are sorry for the lack of clarity. Mantel correlograms were applied to detect phylogenetic signal in order to apply the method of community assembly using phylogenetic metrics. The purpose of the method is rephrased in the M&M section. It now reads:

L184-L186: *'Mantel correlograms were applied to detect phylogenetic signals, which correlate the phylogenetic distances to the niche distances at different distance classes (e.g., 9, 49, further information in supplementary).'*

Additional information is now provided in the **supplementary** material to better explain the calculations regarding the Mantel correlograms, as follows:

L41- L52: *'For example, for PAR we took all the records of a given ASV (this was done for all ASVs) and recorded the PAR of each record, the ASV's abundance in each record, and then found the abundance -weighted mean of PAR. This is the ASV's 'niche value' for PAR. The analogous procedure was used to estimate ASV niche values for all the other environmental parameters. To summarize major trends in this relationship, between-ASVs niche differences were placed in phylogenetic distance bins and median niche difference was found in each bin (which are represented by the squares in the graph). X axis shows the phylogenetic distances (0: lowest, 1.0 maximum). Y axis shows the correlation between environmental optima and phylogenetic distances. All mantel correlograms showed positive correlation between niche optima and phylogenetic distances at short phylogenetic distances, which justifies the utilisation of the phylogenetic metrics to infer community assembly [11].'*

Discussion

L339-347: I am a bit confused here: In the beginning of the paragraph the authors discuss that environmental filtering, a deterministic process, is evidenced for phytoplankton community assembly and dynamics. But then they seem to contradict themselves by mentioning that drift was more important than selection during most of the year. Please clarify.

This is now clarified in the beginning of the discussion, as well as in the abstract, the material and methods. The table 1 was slightly modified for the same reason. For this an additional sentence was also added in the discussion section.

It now reads:

L346-L347: *'Alpha diversity of phytoplankton communities was regulated by environmental filtering.'*

The following sentence was slightly modified as follows:

L347-349: *'Drift, followed by homogeneous selection, were the major mechanisms regulating the temporal turnover in community composition (beta diversity) and prevailed across seasons.'*

L370-373: I think this is redundant, all these mechanisms could be considered deterministic rather than random. For example, productivity depends on community structure and abundances, which can be driven by environmental filtering as shown.

We agree, all these mechanisms could be considered deterministic, however as it has been shown in the studies that are referenced in the text, these mechanisms can have a stochastic effect in the communities. An example is now added to illustrate this better as follows:

L432-L434: *'For example, predators can increase the importance of stochastic processes by reducing the number of individuals that can live in a given environment, and thus the community size, by increasing the probability of species going extinct locally (77).'*

L403: Is *P. pungens* a mixotroph? According to the listed references or my knowledge, I couldn't confirm this, although in Burkholder et al it is mentioned that *P. australis* can employ osmotrophy.

Indeed, we couldn't confirm this either. It is now removed (**L464**)

L441-443 are repeating the same text as few lines above. You could delete L427-428 and change accordingly the text.

We deleted the repetition from the text (initially submitted version: L427-L428)

September 20, 2022

Mx. Dimitra-Ioli Skouroliaiou
Laboratory of Oceanography and Geosciences UMR 8187 CNRS ULCO
28 Av. du Maréchal Foch
Wimereux 62930
France

Re: Spectrum02427-22R1 (Stochastic and deterministic processes regulate phytoplankton assemblages in a temperate coastal ecosystem)

Dear Mx. Dimitra-Ioli Skouroliaiou:

Your manuscript has been accepted, and I am forwarding it to the ASM Journals Department for publication. You will be notified when your proofs are ready to be viewed.

Sincerely,

Konstantinos Kormas
Editor, Microbiology Spectrum
